# Near-Infrared-II Fluorescence Imaging of Tumors with Organic Small-Molecule Fluorophores

**DOI:** 10.3390/s25227080

**Published:** 2025-11-20

**Authors:** Mao Guo, Xiaomu Hu, Wei Du

**Affiliations:** 1New Cornerstone Science Laboratory, MOE Key Laboratory for Analytical Science of Food Safety and Biology, College of Chemistry, Fuzhou University, Fuzhou 350108, China; guomao6136@outlook.com; 2Department of Pharmacy, 900TH Hospital of Joint Logistic Support Force of PLA, Fuzhou 350025, China

**Keywords:** NIR-II, fluorescent probe, organic small-molecule fluorophores, tumor imaging

## Abstract

Over the past decade, near-infrared-II (NIR-II, 1000–1700 nm) fluorescence imaging has become a focal point in tumor imaging due to its advantages of low light scattering, weak biological autofluorescence, extraordinary penetration depth, high signal-to-background ratio, and micron-level high resolution. To date, a large number of NIR-II materials have been developed for tumor imaging. Among them, NIR-II organic small-molecule fluorophores have emerged as research hotspots owing to their distinctive advantages, such as superior optical properties, excellent controllability, favorable biocompatibility, and tunable pharmacokinetics. In this review, we summarize the latest progress in lNIR-II fluorescent probes based on organic small-molecule fluorophores for tumor imaging, focusing on their structural features, design principles of NIR-II fluorescent probes, and applications in tumor imaging. Finally, we will discuss the challenges, future prospects, and development directions of organic small-molecule fluorophores for NIR-II fluorescence imaging of tumors.

## 1. Introduction

Cancer remains a major public health burden characterized by high incidence, mortality, and pronounced inter- and intratumoral heterogeneity. Tumor initiation, progression, and metastasis are tightly regulated by the tumor microenvironment (TME) [1,2,3], a dynamic ecosystem comprising malignant cells, vasculature, immune cells, cancer-associated fibroblasts, diverse signaling molecules, and the extracellular matrix (ECM) [3,4]. The TME exhibits hallmark pathophysiological features, including aberrant vasculature, redox dysregulation (e.g., elevated reactive oxygen species, ROS), hypoxia, acidosis, hyperactivation of specific enzymes (e.g., matrix metalloproteinases), and abnormal expression of molecular biomarkers. These dysregulated entities—such as receptor proteins (e.g., epidermal growth factor receptor (EGFR), human epidermal growth factor receptor 2 (HER2)), tumor-associated antigens (e.g., PSA, CEA), ROS, metabolites, and nucleic acid alterations—constitute tumor-specific biomarkers and therapeutic targets, forming the foundation for molecular imaging and targeted therapy [5,6,7]. Collectively, they drive proliferation, invasion, metastasis, and therapy resistance, while complicating early detection and precise intervention [2,8]. Many cancers are curable when detected early and treated effectively. Thus, technologies for early detection, accurate localization, and real-time visualization are essential to improve diagnosis, guide surgery, assess treatment response, and monitor prognosis [9,10,11,12].

Clinically, commonly used imaging modalities include computed tomography (CT), magnetic resonance imaging (MRI), positron emission tomography (PET), and ultrasonography [13,14,15,16,17]. Each contributes uniquely to tumor diagnosis, staging, and treatment assessment: CT affords high spatial resolution, MRI provides excellent soft-tissue contrast, PET reports metabolic activity, and ultrasound offers real-time, user-friendly operation [15,18,19,20]. Nonetheless, these modalities have notable limitations: limited spatial resolution (e.g., PET), ionizing radiation exposure (e.g., CT and PET), high costs (e.g., MRI and PET), and challenges in achieving long-term, highly sensitive real-time imaging [17,19,20,21,22]. Against this backdrop, optical imaging—particularly fluorescence imaging—has gained prominence in basic and translational oncology owing to its high sensitivity, high spatiotemporal resolution, absence of ionizing radiation, and capability for multiwavelength real-time imaging [23,24,25]. Fluorescence imaging allows precise tumor localization and margin delineation, and with, specific molecular probes, reveals TME physiology and molecular signatures, thereby aiding early diagnosis, intraoperative navigation, and therapy monitoring [26,27,28,29].

Within fluorescence imaging, near-infrared-I (NIR-I, 700–900 nm) imaging has achieved meaningful advances in preclinical research and selected clinical applications [30,31,32,33]. However, strong tissue scattering and autofluorescence within this spectral window constrain penetration depth (<1 cm), diminish the signal-to-noise ratio (SNR), and limit spatial resolution, thereby hindering deep-tissue imaging [34,35]. By contrast, fluorescence imaging in the second near-infrared window (NIR-II, 1000–1700 nm) has rapidly emerged as a highly promising in vivo modality due to markedly reduced tissue scattering, minimal autofluorescence, and lower photon attenuation [36,37,38,39]. Compared with NIR-I, NIR-II offers deeper tissue penetration (up to several centimeters) and higher spatiotemporal resolution, enabling high-contrast, micrometer-scale biological imaging [40,41,42,43,44]. However, it is crucial to note that the benefits across the broad NIR-II spectrum are not uniform. Light scattering in biological tissues decreases monotonically with increasing wavelength from 400 to 1700 nm, whereas water absorption exhibits a distinct peak between 1350 and 1600 nm (Figure 1a,b) [45]. Due to the trade-off between reduced scattering and elevated water absorption, the 1000–1350 nm range is particularly favorable for deep-tissue imaging, offering lower scattering and relatively low water absorption, compared to other NIR-II regions.

A broad spectrum of emitters has been developed for NIR-II imaging, including carbon nanotubes [46,47], quantum dots [48,49,50], lanthanide-doped nanoparticles [51,52,53], and a variety of organic materials such as conjugated polymer nanoparticles (CPNs) and small-molecule fluorophores [54,55,56,57]. The potential long-term biotoxicity and metabolic retention of inorganic nanomaterials, however, substantially impede their clinical translation. In contrast, CPNs and small-molecule fluorophores offer obvious advantages, such as biocompatibility and tunable optical properties through chemical structure modulation. However, CPNs often suffer from slow metabolism and poor water solubility when used for in vivo imaging. In terms of small-molecule fluorophores, they exhibit faster and more predictable clearance than inorganic nanomaterials and CPNs, making them among the most promising NIR-II imaging agents for clinical use [58,59,60,61,62]. A notable clinical milestone was reported in 2020, when Tian et al. successfully performed NIR-II fluorescence-guided surgery in 23 patients with primary and metastatic liver cancer using indocyanine green (ICG), marking a critical step toward clinical translation of organic NIR-II fluorescent probes [63].

This review systematically summarizes recent advances in NIR-II fluorescent probes based on organic small-molecule fluorophores for tumor imaging, with emphasis on molecular structural features, rational design strategies, and applications in tumor diagnosis, real-time intraoperative navigation, and therapy-response monitoring. We conclude by discussing key challenges and future directions in this rapidly evolving field.

## 2. NIR-II Organic Small-Molecule Fluorophores

Substantial progress has been made in the molecular design of small-molecule fluorophores with emission in the NIR-II region. Current efforts predominantly focus on four structural archetypes: donor–acceptor–donor (D-A-D) frameworks, cyanine derivatives, boron-dipyrromethene (BODIPY) derivatives, and xanthene dyes [64,65]. These architectures afford tunable photophysical properties and strong potential for biomedical applications. Recent strategies focus on rational structural optimization to improve optical performance, achieving higher fluorescence quantum yield (QY), greater photostability, and better aqueous solubility. Such functionally refined probes are increasingly used for high-contrast in vivo tumor imaging and pathophysiological visualization [66,67]. Some representative NIR-II organic small-molecule fluorophores and their photophysical properties have been listed and summarized in Table 1.

### 2.1. Donor–Acceptor–Donor NIR-II Small-Molecule Fluorophores

D-A-D fluorophores generally comprise an electron-accepting (A) core symmetrically connected to two electron-donating units via π-conjugated linkers (Figure 2). Among reported scaffolds, benzobisthiadiazole (BBTD) is a prototypical acceptor due to its strong electron-withdrawing character, narrow bandgap, high molar extinction coefficient, long-wavelength emission, structural tunability, and pronounced intramolecular charge-transfer (ICT) features, collectively conferring excellent photostability [65,76,77]. Alternative acceptors recently explored include [1,2,5]thiadiazolo[3,4-f]benzotriazole (TBZ) [78], thienoisoindigo (TIIG) [79], 6,7-bis(4-hexyloxyphenyl)-4,9-di(thiophen-2-yl)[1,2,5]thiadiazolo[3,4-g]quinoxaline (TTQ) [79,80], [1,2,5]thiadiazolo[3,4-g]quinoxaline (TQ), thienothiadiazole (TTD) [81,82,83], and 6,7-dithienyl [1,2,5]thiadiazolo[3,4-g]quinoxaline (TQT) [84]. Donor units commonly incorporate phenyl-based motifs such as triphenylamine (TPA), tetraphenylethylene (TPE), alkylfluorenes, or alkylbenzenes, while thiophene derivatives frequently serve as π-bridges [85].

Strategic molecular engineering tunes the electron-donating/withdrawing strengths of donor and acceptor components, thereby modulating the highest occupied molecular orbital–lowest unoccupied molecular orbital (HOMO–LUMO) gap and precisely adjusting NIR-II absorption/emission profiles [86,87,88]. Common approaches to enhance fluorescence quantum yield include: (1) Incorporation of steric-shielding units to suppress nonradiative decay [89,90]; (2) inhibition of twisted intramolecular charge-transfer (TICT) states [91]; (3) confinement within hydrophobic microenvironments to reduce solvent-induced quenching [92]; (4) construction of aggregation-induced emission (AIE)-active scaffolds [93,94]. Owing to their well-defined structures, tunable photophlysics, favorable metabolism, and excellent biocompatibility, D-A-D small molecules have emerged as pivotal candidates for organic NIR-II probes, driving intense research interest.

### 2.2. NIR-II Cyanine Fluorophores

Cyanine fluorophores feature a central polymethine chain [(CH)ₙ] conjugated to terminal heterocycles containing nitrogen, sulfur, or oxygen (Figure 3) [95]. Typically, one positively charged heterocycle acts as an electron acceptor, while the opposing heterocycle serves as an electron donor, producing an intrinsic push–pull effect that extends π-conjugation. NIR-II emission commonly arises from ICT facilitated by excited-state structural distortion: upon excitation, elongation of central C–C bonds increase rotational freedom, perturbs backbone symmetry, and activates long-wavelength emission [85,96].

Bathochromic shifts in absorption/emission can be achieved by polymethine chain elongation, donor–acceptor modulation (e.g., sulfur-for-oxygen substitution), or J-aggregation, thereby improving penetration depth and spatial resolution in tissues [71,97,98]. Photostability and quantum yield can be enhanced by installing electron-withdrawing groups or steric hindrance along the polymethine bridge, and by forming protein complexes [70,74,99]. To overcome hydrophobicity and improve biocompatibility, two main strategies are employed: (1) Nanoprecipitation with amphiphilic polymers to encapsulate the dye, and (2) molecular engineering with hydrophilic substituents such as sulfonates or Poly(ethylene glycol) (PEG) chains [95,100]. With high molar extinction coefficients, tunable emissions, synthetic versatility, and modularity, cyanine fluorophores hold substantial promise for biomedical imaging and have been widely applied for in vivo tumor visualization.

### 2.3. NIR-II BODIPY Fluorophores

BODIPY dyes are distinguished by high fluorescence quantum yields, narrow emission bandwidths, robust photostability, synthetic versatility, minimal autofluorescence, low cytotoxicity, and favorable biocompatibility (Figure 4) [101,102,103]. The rigid, planar, highly conjugated core consists of two pyrrole rings chelated to a BF_2_ unit, enabling extensive site-specific modification. Based on core variation, BODIPY derivatives are broadly categorized as conventional BODIPYs or aza-BODIPYs [66].

Aza BODIPYs, formed by isoelectronic substitution of the meso carbon of BODIPYs with nitrogen, exhibit reduced HOMO–LUMO gaps and bathochromic extending into the NIR-II region [85,104], broadening their utility in deep tissue imaging and phototherapy. Hydrophilic functionalization at the 2,6-positions, combined with quaternary ammonium groups on boron, markedly improves aqueous solubility, obviating the need for extrinsic encapsulation or PEGylation [100,105,106]. The electron-deficient aza-BODIPY scaffold also supports D-A-D construction via donor installation at the 3,5- or 1,7-positions, promoting strong ICT and enabling NIR-II emission [85,107]. The inherent synthetic flexibility of the BODIPY core facilitates incorporation of functional moieties for multimodal theranostics, significantly advancing tumor-targeted precision imaging.

### 2.4. NIR-II Xanthene Fluorophores

Xanthene dyes possess a tricyclic core in which two benzene rings are bridged by an oxygen atom and a methylene group, forming a six-membered heterocycle (Figure 5). This architecture affords high molar extinction coefficients, superior photostability, low cytotoxicity, high fluorescence quantum yields, facile functionalization, and tunable emission. Representative derivatives include rhodamine, fluoresceine, and related analogs [60,66,108].

Extensive efforts have targeted bathochromic shifts through π-extension strategies that expand conjugation within the xanthene scaffold. Covalent fusion of naphthalene rings (fused polycyclic aromatics) or installation of styryl groups (rigid π-bridges) effectively extends π-systems and red-shifts absorption/emission [66,75,109]. Heteroatom substitution of the central oxygen with silicon or sulfur narrows the HOMO–LUMO gap, yielding further red-shifts. Increasing molecular rigidity via planarization suppresses rotation about nitrogen centers and mitigates TICT. Incorporation of julolidine or tetrahydroquinoline (THQ) substituents enhances quantum yield and can produce large Stokes shifts by strengthening ICT [28,108].

## 3. Design Strategy of NIR-II Fluorescent Probes Based on Organic Small-Molecule Fluorophores for Tumor Imaging

Based on the signal generation mechanism, there are two major strategies for designing NIR-II fluorescent probes for tumor imaging using organic small-molecule fluorophores (Figure 6): always-on fluorescence imaging strategy and activatable fluorescence imaging. “Turn-off” probes (for example, scaffolds quenched until biomarker interaction) are not considered here due to their susceptibility to environmental perturbations (e.g., temperature- or oxygen-induced false quenching) and their typically low SNRs [110,111]. This section focuses on probes that either exhibit intrinsically stable emission or undergo tumor-specific activation, thereby enabling reliable in vivo imaging contrast with deep-tissue penetration.

### 3.1. Always-On Fluorescence Imaging

Always-on fluorescence imaging strategy employs probes that maintain fluorescent throughout systemic circulation. This strategy uses tumor-targeting moieties to selectively accumulate signaling agents in tumors, creating concentration differences between malignant and healthy tissues and enabling lesion visualization through differential signal intensity. Its key advantages—no need for external activation, real-time monitoring, and predictable signal output—support intraoperative navigation. However, its imaging performance is tightly coupled to delivery efficiency, typically achieved via active or passive targeting [112,113,114]. Typically, always-on fluorescence imaging of tumors can be achieved either by passive targeting strategy or active targeting strategy (Figure 6a).

For passive targeting, hydrophobic dyes are always encapsulated within nanocarriers (e.g., DSPE-PEG micelles, DOPC liposomes, or human serum albumin (HSA)) to exploit the enhanced permeability and retention (EPR) effect in solid tumors. Rapid angiogenesis produces leaky vasculature that permits macromolecules (approximately 100–800 nm in diameter) to extravasate and accumulate in tumor tissue; concurrently, impaired lymphatic drainage promotes retention [115]. Thus, the size of passive targeting probes is always tuned to the nanometric range (typically 20–200 nm) to promote selective accumulation in tumor tissue. This strategy offers broad applicability and prolonged circulation, but may suffer from modest tumor accumulation and nonspecific uptake by the liver and spleen [116].

For active targeting, fluorophores are covalently conjugated to tumor-targeting moieties (antibodies, peptides, or small-molecule ligands). High-affinity ligand–receptor interactions or antigen–antibody recognition facilitates efficient fluorophore binding to the membrane of tumor cells [117,118,119,120,121]. Continued optimization of hydrophilicity, biocompatibility, and size will further enhance targeted accumulation and penetration–retention in vivo [114]. For example, flexible PEG linkers can help alleviate steric hindrance and minimize self-quenching, thereby improving specificity and accelerating tumor enrichment [112,122,123].

### 3.2. Activatable Fluorescence Imaging

Unlike always-on fluorescence imaging, activatable fluorescence imaging strategy reduces dependence on absolute probe concentration and enable more precise lesion identification [114,124,125]. A canonical activatable probe comprises three elements: a fluorophore, a linker, and a recognition moiety [126]. Commonly used recognition moieties include protease-cleavable peptide linkers (e.g., for MMP-2/9 or caspase-3) paired with quenchers (e.g., BHQ-3), pH-sensitive tertiary amines, and units responsive to redox species like ROS, reactive nitrogen species (RNS), and reactive sulfur species (RSS) [127,128,129]. Activatable fluorescent probes remain in the fluorescence “off” state until triggered by tumor-associated stimuli such as enzymes, ROS, or acidic pH, at which point they switch to an emissive (“on”) state. Activatable fluorescent probes offer ultralow background interference and microenvironment-responsive specificity, thereby improving sensitivity and selectivity. NIR-II activatable probes support noninvasive, real-time monitoring of tumor biomarkers and microenvironmental cues and are commonly implemented as single-channel or ratiometric systems (Figure 6b).

Single-channel probes recognize tumors through an on–off switching mechanism within a single emission channel upon response to specific tumor-associated stimuli [127,128]. Despite their utility, single-channel probes can be influenced by probe distribution and tissue-dependent attenuation, complicating quantitative analysis [130]. By contrast, ratiometric activatable probes provide dual-signal self-calibration to minimize variability related to instruments and environmental factors, thereby enabling more accurate semi-quantitative and quantitative measurements [114,131]. Generally, there are two different types of a ratiometric activatable fluorescence imaging strategies (Figure 6b). (1) Dual responsive signals, where a probe generates two interconnected outputs—one enhanced and one reduced upon responding to the target stimulus, enabling reversible ratiometric detection through wavelength-intensity ratios [131,132]. (2) Responsive signal with an internal reference, in which a target-sensitive channel is combined with a stable reference channel to allow real-time normalization and reliable ratiometric measurement [133].

## 4. Advances in NIR-II Fluorescent Probes Based on Organic Small-Molecule Fluorophores for Tumor Imaging

The development of new NIR-II organic small-molecule fluorophores and advanced molecular design strategies has established NIR-II fluorescent probes with deeper tissue penetration and higher spatiotemporal resolution, conferring distinct advantages for tumor imaging. This section highlights representative studies of NIR-II fluorescent probes based on organic small-molecule fluorophores, aiming to inform and guide the design and application of next-generation fluorescent probes for tumor imaging.

### 4.1. Always-On NIR-II Fluorescent Probes

The unique structural and metabolic features of tumors (e.g., aberrant angiogenesis and impaired lymphatic drainage) provide a biological basis for targeted probe design. Recent reported always-on NIR-II fluorescent probes have adopted either a passive targeting strategy or an active targeting strategy to improve targeting ability toward malignant tissues [134,135].

#### 4.1.1. Passive Targeting Probes

NIR-II organic small-molecule fluorophores typically exhibit limited aqueous solubility and poor tumor-targeting capability. To confer passive tumor-targeting capability to these fluorophores, chemical modification and nanocarrier encapsulation represent the two primary strategies.

Chemical modification can directly optimize the physicochemical properties of NIR-II organic small-molecule fluorophores (e.g., solubility, hydrodynamic size, and surface charge), thereby prolonging circulation, reducing carrier dependence, and enhancing EPR effect utilization. PEGylation is a well-established approach to amplify the EPR effect by extending blood circulation and improving vascular permeability [136,137]. The hydrophilic PEG layer reduces opsonization and protein adsorption, thereby minimizing liver and spleen clearance. It also masks surface charge, weakens electrostatic interactions with healthy endothelium, and adjusts hydrodynamic diameter to promote extravasation.

Zhang and co-workers introduced PEG chains onto an aza-BODIPY core to construct a brush-like macromolecular probe, FBP 912 (**1**) (Figure 7a) [138]. This polymer brush exhibits a prolonged circulation half-life (t_1_/_2_ ≈ 6.1 h); its brush architecture effectively suppresses aqueous-phase quenching, yielding substantially higher NIR-II brightness than that of a conventionally PEGylated, renally clearable control fluorophore **1a** (t_1_/_2_ < 2 h). After tumor-bearing mice were intravenously (i.v.) injected with **1** and subjected to NIR-II fluorescence imaging, the tumor-to-background ratio (TBR) of **1** reached 8.2 ± 0.5 at 24 h, exceeding previous renally clearable probes and showing effective passive targeting imaging of subcutaneous tumors (Figure 7b–f).

PEGylation of other NIR-II fluorophores can also endow them with passive tumor-targeting capability. Fan and colleagues appended four hydrophilic PEG chains to a selenium-oxanthene (SO)-based NIR-II emissive scaffold and obtained a water-soluble probe, PSO (**2**) (Figure 8a) [139]. Under passive targeting, **2** enabled clear tumor imaging as early as 4 h post-dosing (with a 5.4-fold enhancement in signal relative to surrounding tissue) and maintained strong signals at 28 h, indicating excellent NIR-II performance (Figure 8b,c). In addition, **2** exhibited favorable renal clearance (Figure 8c).

Furthermore, covalently attaching hydrophilic groups to hydrophobic dyes creates amphiphilic structures that self-assemble into stable nanoaggregates in water; precise size control within the EPR window further promotes passive targeting [140,141]. For example, Li et al. modified the small-molecule NIR-II fluorophore Flav7 with an amphiphilic peptide and constructed probe PF (**3**), which self-assembled into uniform micelles (i.e., **3-NPs**) in water with a particle size of about 90 nm and a surface potential of −12.5 mV (Figure 9a) [142]. **3-NPs** accumulated at tumor sites via the combined effects of EPR and nanomaterial-induced endothelial leakiness (*NanoEL*). As shown in Figure 9b, within 24 h of i.v. administration of **3-NPs**, tumor fluorescence intensity and TBR continuously increased, indicating the excellent tumor-targeted imaging performance of **3-NPs**.

Besides PEGylation and self-assembly, nanocarrier encapsulation is also a widely used strategy to endow small-molecule NIR-II fluorophores with the EPR effect [143,144,145]. Common carriers include phospholipids (liposomes), polymeric systems (e.g., PEG-PLGA, PLA micelles), and albumin. Encapsulation protects dyes from rapid degradation and clearance, improves in vivo stability, and, owing to the nanoscopic size, significantly prolongs blood circulation to achieve efficient passive enrichment at tumors or other lesions, thereby enhancing imaging and therapeutic outcomes. For example, Li and co-workers reported an ICG-like NIR-II fluorophore, ICR-Qu (**4**), and encapsulate it with DSPE-PEG_2000_ to form **4-NP** (Figure 10a) [146]. **4-NP** possessed high photostability and photothermal conversion efficiency (PCE, 81.1%). After intratumoral administration of **4-NP** into 4T1 tumor-bearing BALB/c mice, strong NIR-II fluorescence signals were observed in the tumor region, from 1 to 6 h post-injection, with significant fluorescence persisting at 48 h (Figure 10b). Fluorescence imaging-guided tumor resection enabled clear delineation of excised tumors and residual lesions, indicating the high sensitivity of **4-NP** for tumor imaging and intraoperative navigation (Figure 10c).

Pluronic F127 (PEO_100_-PPO_65_-PEO_100_) is also a commonly used polymer for the encapsulation of small-molecule NIR-II fluorophores. It is a prototypical triblock copolymer whose hydrophobic PPO forms the micellar core and hydrophilic PEO forms the corona [147]. Wang et al. designed an A–D–A′–D–A dye, SM-16 (**5**), and formulated it with Pluronic F127 to obtain SM 16 NPs (**5-NPs**) as a new NIR-II fluorescent probe (Figure 11a) [148]. Following i.v. injection into 4T1-tumor-bearing mice, **5-NPs** produced tumor-localized fluorescence substantially stronger than surrounding tissues under EPR guidance, with peak contrast at approximately 16 h post-dosing (Figure 11b).

In addition, passive tumor targeting can also be achieved through protein-dye complexes formed with albumin or components of fetal bovine serum (FBS) via hydrophobic interactions or covalent bonds [149,150]. Zhu et al. assembled *β*-lactoglobulin (*β*-LG, 18.4 kDa) with the cyanine dye IR-780 to construct a protein@dye probe, *β*-LG@IR-780 (**6**) (Figure 12a) [151]. This probe exhibited excellent NIR-II brightness, photostability, and favorable biosafety. When **6** was applied in a murine lymphatic metastasis model, NIR-II lymphangiography detected pronounced lymph node enlargement by day 5 post-tumor inoculation, and monitored its gradual deterioration over time (Figure 12b). The inoculated side showed clear lymphatic dilation and faster lymph node visualization than the contralateral side, enabling early detection of tumor-induced lymph node enlargement. Zhang and co-workers combined FBS with the NIR-II J-aggregate dye CL4 (**7**) to form the CL4/FBS (**7/FBS**) complex (Figure 13a) [152]. This complex exhibited maximal fluorescence enhancement at 1235 nm, uniform particle size, suitable zeta potential, and excellent photo- and pH stability, all of which are conducive to EPR-mediated tumor NIR-II fluorescence imaging in vivo (Figure 13b,c).

#### 4.1.2. Active Targeting Probes

Compared with passive targeting probes, NIR-II fluorescent probes based on active-targeting strategies generally exhibit superior tumor accumulation and imaging contrast [121]. The targeting ligands used for tumor NIR-II imaging primarily include four classes: peptide-based motifs, antibodies and their fragments, small-molecule ligands and inhibitors, and biomimetic targeting motifs. Representative advances and key performance metrics are summarized below.

Peptide ligands achieve specificity by recognizing receptors or adhesion molecules overexpressed on tumor cell surfaces (e.g., integrin *α*_v_*β*_3_). To date, various tumor-targeting peptides have been used to modify NIR-II organic small-molecule fluorophores to endow fluorescent probes with active targeting ability. For example, Cao et al. identified a novel peptide, Herceptide, that binds specifically to HER2 [153]. They conjugated Herceptide with ICG to yielded ICG–Herceptide (**8**), enabling NIR-II fluorescence imaging of HER2-positive tumors with a peak tumor-to-normal tissue (T/N) ratio of 7.3 and high-contrast intraoperative guidance in SKOV3 subcutaneous xenografts (Figure 14a–c). Wang and colleagues developed the optical probe IRDye800–RM26 (**9**) targeting the gastrin-releasing peptide receptor (GRPR) for NIR-II fluorescence imaging and intraoperative guidance in malignant brain tumors (Figure 15a,b) [154].

To further enhance tumor specificity, researchers often combine passive and active targeting by integrating nanotechnologies with active targeting warhead. The obtained nanoparticles thus integrate ligand-mediated recognition with EPR-driven enrichment [78]. RGD peptide is a commonly used hydrophilic tumor-targeting warhead that binds to integrin *α*_v_*β*_3_ with high affinity and promotes receptor-mediated endocytosis. Conjugating RGD with a hydrophobic NIR-II organic small-molecule fluorophore always yields amphiphilic molecules that tend to self-assemble into nanostructures with both passive and active tumor-targeting abilities. Zhao et al. used a NIR-II-emissive heptamethine cyanine scaffold and dual-site conjugated cRGD to obtain QT–RGD (**10**), which self-assembled into nanoparticles (Figure 16a) [155]. In a 4T1 mouse model, tumor fluorescence peaked at approximately 4 h after injection of **10**, with a high signal-to-background ratio (SBR) of 8.2, whereas the control probe (i.e., MT) without RGD modification showed only weak signals (Figure 16b,c).

However, dye self-assembly often reduces fluorescence intensity due to the aggregation-caused quenching (ACQ) effect. To suppress self-assembly of RGD-modified fluorophores, incorporating a more hydrophilic linker, such as PEG, is an effective approach. Lyu et al. developed the small-molecule NIR-IIa probe IR–32p (**11**) targeting *α*_v_*β*_3_ integrin by connecting an IR-dye and RGD with a PEG linker (Figure 17a) [156]. After i.v. injection of **11** into an orthotopic glioma model for NIR-II fluorescence imaging, peak T/N ratios were achieved at 8 h, reaching about 4.1 and 6.9 under 808 nm and 1064 nm excitation, respectively (Figure 17b). Notably, after scalp removal, T/N increased to ~9.4 under 1064 nm excitation (Figure 17c). Ex vivo imaging further corroborated *α*_v_*β*_3_ specificity of **111** (Figure 17d).

Besides self-assembly, combined passive and active targeting can also be achieved by loading NIR-II organic small-molecule fluorophores with nanocarriers and further modified with active targeting warhead. Yang et al. prepared a biodegradable nanoprobe, R&HV–Gd@ICG (**12**), by loading ICG into a hollow virus-like gadolinium nanoparticle and modifying the nanoparticle surface with cyclic RGD pentapeptide (cRGDfK), enabling enhanced in vivo imaging of breast cancer (Figure 18a) [157]. In 4T1-tumor-bearing mice, compared with free ICG and the unmodified control probe (HV-Gd@ICG), **12** achieved the highest NIR-II fluorescence intensity, as well as the highest TBR of 5.7 ± 1.0 at 48 h, indicating its excellent tumor-targeted imaging ability (Figure 18b–d).

Antibody-based targeting is also a widely used active targeting strategy. It exploits the high-affinity binding of antibodies to tumor-specific antigens (e.g., bevacizumab against VEGF [158], cetuximab against EGFR [159]) to enable tumor-targeted imaging. To date, several ICG-antibody conjugates for tumor-targeted NIR-II fluorescence imaging have been reported [160,161,162]. Notably, the recombinant monoclonal antibody cetuximab (Erbitux^®^), a Food and Drug Administration (FDA)-approved drug, has been reported for use in NIR-II fluorescence imaging [163]. Zeng et al. synthesized the water-soluble small-molecule NIR-II dye H2a–4T (**13**) and complexed it with cetuximab to form **13@Cetuximab** (Figure 19a) [69]. In an HCT116 colorectal cancer model, the complex clearly delineated tumors between 2–20 h, with a T/N of 5.51 ± 0.12 at 4 h (Figure 19b,c). In comparison, the free dye (**13**) produced substantially weaker tumor signals at all time points (Figure 19b,c). These results demonstrated the efficient active targeting performance of **13@Cetuximab** to EGFR-overexpressing tumors.

Small-molecule ligand-based tumor targeting enables directed probe accumulation at tumor sites via metabolism- or receptor-mediated mechanisms [164]. For instance, folic acid (FA) targets folate receptors overexpressed in various tumors (e.g., ovarian and breast cancers) [165]. Shi et al. used a FA-ICG conjugate for in vivo NIR-II/NIR-I fluorescence imaging of orthotopic U87MG human GBM tumors [166]. Interestingly, the NIR-II fluorescence signal offered superior contrast at all time points compared to the NIR-I fluorescence signal.

Besides receptor-mediated mechanisms, small-molecule ligand-antigen interaction-based targeting is also an effective tumor-targeting strategy, such as the inhibition effect between (S)-5-amino-1-carboxypentyl)carbamoyl)-*L*-glutamic acid (Glu-urea-Lys) and prostate-specific membrane antigen (PSMA). Cui et al. designed PSMA–1092 (**14**), which contained two Glu–urea–Lys motifs with an inhibition constant (K_i_) of 80 pM (Figure 20a) [167]. After tail vein injection of **14** into LNCaP tumor-bearing mice, tumor margins were clearly defined at 2–4 h by NIR-II fluorescence imaging, with peak tumor uptake occurring at 6 h and a maximum T/N of 7.62 ± 1.05 achieved at 12 h (Figure 20b,c). These imaging results clearly demonstrated the targeting capability of compound **14** for tumors.

Small molecule–organelle interaction can also be utilized for tumor targeting. Li et al. introduced the lipophilic cation triphenylphosphine (TPP) into a donor–acceptor (D–A) NIR-II organic small-molecule fluorophore and constructed the mitochondria-targeted probe TQPTPP (**15**) (Figure 21a) [168]. When applying the probe for in vivo NIR-II fluorescence imaging of 4T1 tumors, the probe exhibited prolonged tumor retention, with maximal signal observed on day 5 and an SBR of about 9 on day 6, providing an effective tool for subcellular-level active targeting in NIR-II fluorescence imaging (Figure 21b).

By cloaking probes with endogenous biological membranes (e.g., red blood cells, platelets, macrophages, stem cells, or tumor cell membranes) or natural structures, biomimetic modifications can impart tumor tropism while reducing immunogenicity and improving biocompatibility [169,170,171]. Jiang et al. developed LZ-1105@HAm (**16@HAm**), an albumin–NIR-II cyanine dye assembly coated with A549 tumor cell membranes (Figure 22a) [172]. Homologous recognition by A549-derived membranes markedly enhanced tumor accumulation and NIR-II fluorescence signals in the tumor, reaching maximal uptake at 2 h post-injection, with fluorescence intensity approximately 4.7-fold higher than that of the uncoated control (i.e., **16@HSA**) (Figure 22b). Wei et al. self-assembled the NIR cyanine dye Sulfo-1100 (**17**) with holo-transferrin to form hT-Sulfo-1100 nanoparticles (**hT-17 NPs**) (Figure 23a) [173]. This platform crossed the blood–brain barrier and targeted glioma tumor, yielding strong NIR-II fluorescence signals at orthotopic gliomas as early as 10 min post-i.v. injection into an orthotopic glioma model (Figure 23b). In comparison, **17** alone showed no effective tumor signal in the orthotopic glioma model, demonstrating excellent deep-tissue and transcranial imaging capabilities of **hT-17 NPs**.

Active-targeting NIR-II fluorescent probes based on organic small-molecule fluorophores—spanning peptides, antibodies, small molecules, and biomimetic designs—enable precise recognition of tumor-associated receptors and microenvironments, with marked advances in in vivo imaging contrast, spatiotemporal resolution, and intraoperative guidance. The broadly applicable RGD–*α*_v_*β*_3_ axis, in particular, serves as a key design module that enhances imaging quality and therapeutic gains across multiple solid tumors. Looking ahead, integrating multi-receptor co-targeting, controllable release, and clinically translatable formulation engineering is expected to accelerate the trajectory from animal validation to clinical application.

### 4.2. Activatable NIR-II Fluorescent Probes

Recent advances in small-molecule fluorophore-based NIR-II probes aim to improve detection accuracy, biocompatibility, stability, photobleaching resistance, deep-tissue imaging, and clinical translation. Activatable small-molecule NIR-II probes leverage robust photophysical switching mechanisms to translate TME hallmarks into high-contrast optical readouts. By leveraging TME-specific biochemical cues, activatable probes suppress background fluorescence and reduce nonspecific signals, thereby enabling precise, real-time tumor visualization [174,175,176].

Activatable NIR-II fluorescent probes are typically created by attaching a stimulus-responsive group to a NIR-II organic small-molecule fluorophore, quenching fluorescence through mechanisms such as ICT, TICT, photo-induced electron transfer (PeT), or fluorescence resonance energy transfer (FRET) (Figure 24a–d). Target engagement in the TME induces changes in the probe’s fluorescence properties—typically fluorescence intensity, emission wavelength, or lifetime—thus reporting the presence and dynamics of tumor-associated analytes.

ICT-based activatable probes typically use the recognition unit as a caging group directly linked to the fluorophore, modulating donor–acceptor “push–pull” electron flow. Target-induced cleavage or transformation restores ICT, causing spectral shifts and increased fluorescence [177,178]. For TICT-based activatable probes, donor and acceptor groups adopt a twisted geometry to form a charge-transfer state that typically relaxes via red-shifted emission or non-radiative decay, quenching fluorescence [179,180]. Activation that restricts molecular motion (for example, higher viscosity) suppresses TICT and turns on emission. When the frontier orbitals of the receptor are located between the HOMO and the LUMO of the fluorophore, PeT effect will occur to quench fluorescence [181,182,183]. Target-induced structural or electronic changes disrupt PeT, thereby restoring fluorescence. In FRET-based activatable probes, non-radiative energy transfer from a donor to an acceptor (fluorophore or quencher) suppresses donor emission [184,185,186]. Target recognition that disrupts donor–acceptor coupling activates the fluorescent signal.

Leveraging these mechanisms, diverse activatable NIR-II fluorescent probes based on NIR-II organic small-molecule fluorophores have been rationally designed for tumor imaging. By readout mode, they can be broadly categorized into single-channel probes and ratiometric (self-calibrating) probes.

#### 4.2.1. Single-Channel Probes

Single-channel probes are programmed to respond to overexpressed tumor biomarkers and, upon activation, produce an intensified fluorescence signal suitable for tracking and delineating tumors. They have been widely applied to image ROS, reactive RNS, biothiols, pH, viscosity, and enzymes in real time [125]. Dysregulated redox homeostasis, distinctive TME features, and aberrant enzyme expression serve as orthogonal “keys” to unlock fluorescence, enabling high-contrast, spatiotemporally resolved tumor visualization in vivo.

##### Redox-Activated Probes

Redox balance is fundamental to cellular integrity and is mediated by reactive species networks—ROS (for example, O_2_·^−^, H_2_O_2_), RNS (for example, NO, ONOO^−^), and RSS (for example, H_2_S, glutathione (GSH))—whose dysregulation is implicated in cancer pathogenesis and progression [187,188,189].

ROS and RNS are always used as the targeted biomarker for tumor imaging. For instance, Li and colleagues reported H_2_O_2_-D4 (**18**), a NIR-II fluorescent probe featuring a highly specific turn-on response to H_2_O_2_ (Figure 25a) [190]. Real-time NIR-II fluorescence imaging of a 4T1 tumor xenograft model using **18** revealed a pronounced tumor-localized signal increase, reaching approximately 50-fold above background at 7 h post-administration, enabling clear tumor delineation (Figure 25b,c). An NO-responsive NIR-II fluorescence/photoacoustic probe (DNPS) was developed by encapsulating DNO (**19**) into DSPE-PEG_2000_ micelles [191]. Upon NO exposure, donor nitrosation converts **19** into a benzotriazole, suppressing PeT effect and activating NIR-II fluorescence (Figure 26a). DNPS was then successfully applied to differentiate NO expression between diabetic and non-diabetic breast cancer models, exhibiting a 2.3-fold stronger and more persistent signal in diabetic tumors (Figure 26b,c).

Another type of biomarker reflecting tumor redox balance is reactive sulfur species, which can also serve as a stimulus for activated NIR-II fluorescence imaging of tumors. Lin et al. developed LET-7 (**20**) (Figure 27a), the first GSH-activatable NIR-II fluorescent probe for selective in vivo GSH visualization [192]. After injection of **20** into 4T1 tumor-bearing mice, NIR-II fluorescence signals in the tumor region rapidly and selectively increased within 60 min, clearly delineating the lesion while normal tissues remained dark (Figure 27b–d).

##### Enzyme-Activated Probes

Many enzymes are upregulated in TEM, enabling enzyme-mediated activation of activatable fluorescent probes. To target the lung cancer biomarker cathepsin B (Cath B), Pu’s group constructed SWIMP (**21**) based on a shortwave-infrared hemicyanine-6 scaffold with three conjugated polymethine units (Figure 28a) [193]. Following intratracheal delivery of **21** into 4T1 lung metastasis murine models, in vivo fluorescence imaging in shortwave infrared window (SWIR, 900–1700 nm) was then conducted. The results showed that the SBRs at lung tumor site reached 15 (ventral) and 9.3 (dorsal) at 8 h, substantially exceeding the Rose criterion and enabling precise detection of lung metastasis (Figure 28b–d). Guo and co-workers synthesized Flavchrom-4 (**22**) for activatable dual-mode imaging of endogenous *β*-gal activity in ovarian tumors (Figure 29a) [194]. Following intratumoral administration of compound **22** in SKOV3 tumor-bearing mouse models, a rapid decrease in NIR-I emission intensity was observed at the tumor site, accompanied by a significant increase in NIR-II emission (Figure 29b–e). The NIR-II signal peaked at 60 min post-injection, exhibiting a 6.3-fold enhancement relative to the pre-injection level, thereby enabling real-time monitoring of enzymatic activity in ovarian tumors (Figure 29d,e).

##### PH-Activated Probes

Weak acidity is an important feature of the TME. Thus, pH can be a useful target for activatable fluorescent probes. Wang et al. developed DTTVBI (**23**) with aggregation-induced emission (AIE) characteristics and formulated **23 NPs** using DSPE-mPEG_2000_ and PLGA, which exhibited pH-switchable NIR-II fluorescence (Figure 30a) [195]. After i.v. injection of **23 NPs** in colorectal cancer patient-derived xenograft (PDX) models, tumor NIR-II signals emerged by 3 h and peaked at 12 h, consistent with EPR-mediated accumulation (Figure 30b). Guided by the strong NIR-II fluorescence signal, synergistic type-I photodynamic therapy (PDT)/photothermal therapy (PTT) achieved substantial tumor ablation.

##### Dual-Activated Probes

Because many biomarkers are not tumor-exclusive, single-trigger probes can be vulnerable to nonspecific activation. Dual- or multi-targeting probes improve specificity by requiring the co-occurrence of two or more spatially co-localized stimuli to activate fluorescence [196].

Xiong’s team developed NSCy-1050 (**24**), a probe that remains quenched in normal tissues but is activated by acidic TME and high intracellular viscosity—two hallmarks of many tumors—producing a strong NIR-II fluorescence signal (Figure 31a) [197]. After intravenous administration of **24** into 4T1 tumor-bearing mice, peak tumor fluorescence occurred at 48 h with a tumor-to-liver ratio (TLR) of 19.5:1, while no significant fluorescence was observed in the phosphate-buffer saline (PBS) group (Figure 31b,c). These results demonstrated the excellent NIR-II fluorescence imaging capability of **24** for tumor imaging in vivo (Figure 31b,c). Kim et al. developed HN-PBA (**25**), which exhibits NIR-II fluorescence activation under acidic pH and high H_2_O_2_ levels (Figure 32a) [198]. When administered intravenously to mice with subcutaneous 4T1 tumors, **25** generated a strong NIR-II fluorescence signal, with a T/N ratio 8.2-fold higher than ICG and 4.0-fold higher than the single-trigger H_2_O_2_-responsive control (HN-PBC) (Figure 32b–e). Intraoperative imaging with **25** enabled accurate identification and resection of primary tumors, achieving a T/N of 24.3 (Figure 32f,g).

Single-channel probes targeting redox species, pH, viscosity, and enzymes have demonstrated sensitive, real-time imaging in diverse tumor models. To further enhance specificity and reduce false positives, dual-/multi-stimuli designs that integrate orthogonal biochemical cues are emerging as a compelling strategy. Continued advances in probe chemistry, formulation, and pharmacokinetic tuning—alongside rigorous, quantitative imaging standards—will be critical to accelerate clinical translation.

#### 4.2.2. Ratiometric Probes

Although activatable NIR-II single-channel probes enable selective “turn-on” of fluorescence signals, their signal intensity is susceptible to local probe concentration, instrument drift, and microenvironment fluctuations, thereby reducing detection reliability and potentially leading to false positive or false negative results [199,200]. Moreover, “always-on” and single-channel activatable NIR-II imaging typically provide qualitative readouts, limiting strict quantification. To address these limitations, ratiometric fluorescence strategies employ two emission channels to self-reference, effectively suppressing extrinsic variability and enabling accurate, quantitative detection of target analytes in complex biological environments [201].

By combining the high SNR, deep penetration, and high-fidelity capabilities of NIR-II fluorescence imaging with ratiometric reliability, various NIR-II ratiometric probes based on organic small-molecule fluorophores have been developed for sensitive in vivo imaging and accurate quantification, showing promising diagnostic potential. Depending on how the ratio is constructed, these probes can be broadly classified as dual-wavelength response probes and internally referenced probes.

##### Dual-Wavelength Response Probes

Dual-wavelength response probes are engineered with well-defined structural response features. Upon activation by tumor-associated biomarkers, they undergo chemical or conformational changes that cause predictable shifts in emission wavelength or intensity—such as blue- or red-shifts with increased or decreased brightness—resulting in a clear change in the fluorescence intensity ratio (FIR) between two channels. The two channels can arise from either two emission bands of a single molecule (single-fluorophore-based NIR-II ratiometric probes) or from two linked components (dual-fluorophore-based NIR-II ratiometric probes), which may be combined via chemical conjugation or co-encapsulation in a nanocarrier.

To construct a single fluorophore-based NIR-II ratiometric probe, Zeng et al. designed CFC-GSH (**26**) for imaging *γ*-Glutamyl transpeptidase (GGT) activity in tumors (Figure 33a) [202]. After intratumoral injection of **26** into a murine hepatocellular carcinoma xenograft model, a strong NIR-I fluorescence signal appeared within 1 min, peaked at 30 min, and remained stable, while the NIR-II signal decreased, causing the FIR to rise over time and plateau by about 30 min (Figure 33b–e). Pre-treatment with the GGT inhibitor acivicin markedly suppressed the ratiometric response of **26** by approximately 19.52-fold, confirming its specificity for GGT (Figure 33b–e). By self-calibration, **26** effectively mitigated interference from endogenous hepatic GGT activity, enabling accurate localization of peritoneal metastases and demonstrating potential for screening and diagnostic imaging.

To enable NIR-II ratiometric fluorescence imaging with longer emission, Lei and colleagues synthesized Rap-N (**27**), based on a polymethine–thiourea scaffold for self-calibrated nitroreductase (NTR) imaging (Figure 34a) [131]. The maximal emission wavelength of **27** was 1010 nm and blue-shifted to 945 nm upon NTR-triggered acylation of its secondary amine, generating a distinct ratiometric signal (Figure 34a). In a 4T1 breast cancer model with in situ injection of **27**, FIR (FL1000LP/FL900LP) of the NTR inhibitor-treated side remained stable, whereas it showed a gradual increase on the contralateral (uninhibited) side within 30 min (Figure 34b). In addition, progressive NIR-II fluorescence enhancement was observed in both 4T1 and CT-26 tumor models after in situ injection of **27**, with a higher FIR in 4T1 tumors than in CT-26 tumors (~0.59 vs. ~0.53), consistent with the more severe hypoxia in 4T1 tumors (Figure 34c).

In addition to single-fluorophore-based NIR-II ratiometric probes, dual-fluorophore-based NIR-II ratiometric probes—where two channels originate from different fluorophores—are also commonly used for tumor imaging. Song et al. developed BBT-IR/Se-MN (**28**), covalently incorporating a ROS-insensitive D-A-D fluorophore (BBT) and a ROS-responsive cyanine dye (IR), enabling FRET from BBT to IR that quenches BBT emission at 1050 nm (Figure 35a) [203]. ROS exposure decomposed IR, disrupting FRET and restoring BBT’s NIR-II fluorescence, leading to a clear increase in the ratiometric signal (FL_1050 nm_/FL_1250 nm_). In an orthotopic U87 glioma mouse model, intravenous administration of **28** yielded immediate, high-contrast NIR-II fluorescence signals for precise tumor localization (Figure 35b). After radiotherapy, the **28**-treated tumors showed significantly higher ratiometric changes compared with the controls in the groups receiving **28** + X-ray or BBT-IR/Se + X-ray (Figure 35b–e). This system enables quantitative ROS monitoring during radiotherapy, reducing tissue-specific confounders and supporting early evaluation of radiosensitivity.

Nanoencapsulation is an effective alternative to covalent linking for incorporating two fluorophores into dual-fluorophore-based NIR-II ratiometric probes. Lin et al. developed a FRET-based H_2_S ratiometric probe (FRHS, **29**), which was fabricated by DSPE-PEG_2000_-assisted encapsulation of an H_2_S-responsive acceptor (LET-1055, **29-1**; NIR-II cyanine scaffold) and an H_2_S-inert donor (Rh930, **29-2**; rhodamine–polymethine hybrid) (Figure 36a) [204]. Upon activation of **29** by H_2_S, its NIR-II fluorescence in Channel 1 (Ch.1, 900–1000 nm, attributed to **29-2**) enhanced, while that in Channel 2 (Ch.2, 1000–1700 nm, attributed to **29-2**) decreased (Figure 36a). After i.v. injection of **29** into orthotopic liver tumor-bearing mice, the tumor signal in Ch.1 markedly increased and that in Ch.2 slightly increased within 12 h post-injection, elevating the Ch.1/Ch.2 ratio (Figure 36c). Notably, tumor-bearing mice exhibited approximately 5.8-fold and 6.3-fold ratiometric enhancements in NIR-II fluorescence signals in the in vivo and ex vivo livers, respectively, compared to healthy mice (Figure 36b,c). These results demonstrated that **29** had the feasibility of in vivo ratiometric NIR-II fluorescence imaging of H_2_S.

Stimuli-triggered FRET formation, like stimuli-triggered FRET disruption, is an effective approach for generating FIR and enables ratiometric fluorescence imaging. Zhang et al. developed a tunable NIR-II ratiometric pH sensor (pTAS, **30**) by encapsulating the NIR-II donor aza-BODIPY (NAB, **30a**) and the pH-responsive rhodamine–hydrazone polymethine pre-acceptor (NRh, **30-b**) into DSPE-PEG_2000_-OCH_3_ micelles (Figure 37a,b) [205]. Under acidic conditions, **30-b** was protonated to form NRhH^+^ (**30-c**), triggering FRET from **30-a** to **30-c** and increasing the FIR (Ch.1/Ch.2) (Figure 37a,b). Following i.v. injection into tumor-bearing mice, both pTAS-2 (**30-2**, 1:5 molar ration of NAB and NRh) and pTAS-3 (**30-3**, 1:10 molar ration of NAB and NRh) accumulated rapidly in tumors within 2 h and persisted for >24 h (Figure 37c–f). Their ratio signals stabilized at around 1.88 and 1.74, corresponding to pH 6.82 and 6.83, respectively, enabling a prolonged and stable imaging window (Figure 37e,f). These ratiometric readouts closely matched microelectrode pH measurements, with coefficients of variation below 0.31% and 0.77%, expanding the measurable range without loss of sensitivity.

Dual-wavelength response NIR-II ratiometric probes establish robust, self-calibrated readouts by translating target-triggered spectral shifts and intensity redistribution into quantitative FIR changes. Representative systems targeting key biomarkers (for example, GGT, NTR, ROS, pH, and ClO^−^) have achieved sensitive, specific, and quantitative imaging across multiple tumor models and have facilitated therapy monitoring and intraoperative decision-making. Future priorities include scalable synthesis, pharmacokinetic optimization, and standardized ratiometric imaging workflows to accelerate clinical translation.

##### Internal Reference Probes

Internal-reference ratiometry uses a target-insensitive reference channel simultaneously with the target-responsive channel to enable real-time normalization. Minimizing spectral overlap between channels reduces crosstalk and improves accuracy and reproducibility of the ratiometric readout. This self-calibration strategy is conceptually straightforward and experimentally accessible, and promising for designing NIR-II ratiometric probes [201,206]. A common approach for constructing internal reference probes combines two optically complementary NIR-II emitters in a single nanoplatform, such as organic–lanthanide hybrids, organic–quantum dot composites, or all-organic assemblies—to ensure co-localization and matched pharmacokinetics in vivo.

Key design principles for robust internal-reference imaging include: (1) Spectral orthogonality—selecting two channels with well-separated excitation and emission to minimize crosstalk; (2) channel roles—combining a stable, target-insensitive reference channel under physiological conditions with a highly specific sensing channel that exhibits a strong dynamic response; (3) colocalization and pharmacokinetic (PK) matching—embedding both emitters in a single carrier to align biodistribution and clearance, enabling reliable self-calibration; (4) readout optimization—using orthogonal excitation/emission configurations or energy-transfer switches to maximize SNR and quantitative robustness. Below are representative internal-reference NIR-II ratiometric probes based on organic small-molecule fluorophores and their key performance features.

Tian and colleagues reported an H_2_S-activatable NIR-II ratiometric nanoprobe (**31**) for in vivo colorectal cancer detection (Figure 38a) [207]. This nanoprobe was constructed by encapsulating a H_2_S-inert aza-BODIPY dye (**31-1**; as the internal reference) and a H_2_S-responsive BODIPY dye (ZX-NIR, **31-2**; for the sensing channel) within the hydrophobic interior of core–shell silica nanocomposites (Figure 38a). Upon injection, **31** produced a strong NIR-II tumor signal in HCT116 tumor-bearing mice within 1 min, stabilizing by 30 min, with a T/N ratio of approximately 5.7 (Figure 38b). The probe selectively identified H_2_S-rich colorectal cancer and, through dual-channel imaging, discriminated cancer types with distinct H_2_S levels (Figure 38b–e).

Formation of energy transfer system is a promising approach to design internal-reference NIR-II ratiometric probes. Song and co-workers designed a NO-responsive, dual-excitation NIR-II ratiometric nanoprobe (DERF-NO, **32**), which was doped with BSBT as FRET acceptor and NO-responsive SIR-NO as FRET pre-donor (Figure 39a) [208]. Upon NO activation, SIR-NO transforms into SIR, triggering an intraparticle FRET system. This increases the NIR-II emission (1000–1700 nm) under 660 nm excitation (FL_660Ex_), while the 808 nm excited signal (FL_808Ex_) remains stable, enabling a reliable dual-excitation ratio (Figure 39a). After i.v. injection of **32** into 4T1 tumor-bearing mice pretreated with PBS or IFN-*γ*, the 660 nm–excited NIR-II signal increased continuously over 24 h in IFN-*γ*–treated tumor models, yielding a FL_660Ex_/FL_808Ex_ ratio of 1.71 at 24 h post-injection of IFN-*γ*—approximately 1.39-fold higher than in PBS controls (Figure 39b). These results indicated the potential of **32** as a promising tool for visualizing and predicting macrophage polarization in tumor.

It is worth noting that in an internal-reference NIR-II ratiometric probe, the fluorophore used as the reference channel does not have to act as the donor or acceptor in the energy transfer system formed upon stimulation. Wu et al. optimized a diene optoelectronic material (1-Br-Et) and encapsulate it with a NIR-I fluorophore NIR775 and a NIR-II fluorophore IR1048 in a micellar nanoparticle to construct a hydroxyl radical (•OH )-responsive NIR ratiometric–photoacoustic (PA) probe (1-NP, **33**) (Figure 40a) [209]. Oxidation of 1-Br-Et by •OH generates 2-Br-Et, which absorbs strongly at 767 nm and quenches the fluorescence of NIRI775 (FL_780_), while minimally affecting the NIR-II fluorescence of IR1048 (FL_1113_); thus, the FL_780_/FL_1113_ ratio decreases significantly (Figure 40a). After i.v. injection of **33** into 4T1 tumor-bearing mice following different radiotherapy doses, the tumor FL_780_ signal decreased significantly in the 10 Gy group, with a normalized ratio (ΔF_780_/ΔF_1113_ = 0.40 ± 0.07) reduced by approximately 2.7-fold, while the 0 Gy group showed minimal change (Figure 40b). These results demonstrated that **33** had the potential to predict tumor response to radiotherapy through •OH-triggered FL_780_/FL_1113_ ratio changes.

Apart from organic small-molecule fluorophores, the fluorophore used as the reference channel can also be inorganic materials. Yang et al. developed a dual-channel NIR-II ratiometric platform (DCNP@IR-806, **34**) constructed by linking down-conversion nanoparticle (DCNP) and the antenna dye IR-806 via a caspase 3-responsive peptide (Figure 41a) [210]. Due to nonradiative energy transfer (NRET) from IR-806 to DCNP in **34**, 1550 nm emission is detected under 808 nm excitation (FL_Ex808,1550_), serving as the sensing channel, while 1550 nm emission under 980 nm excitation (FL_Ex980,1550_) acts as the reference channel for self-calibration (Figure 41a). Upon cleavage of the peptide by caspase-3, the NRET system is disrupted, resulting in a decrease in the FL_Ex808,1550_ signal, while the FL_Ex980,1550_ signal remains stable. Therefore, the FL_Ex980,1550_/FL_Ex808,1550_ ratio can be used to monitor caspase-3 activity in real time. When **34** was applied to an orthotopic hepatocellular carcinoma (OHCC)-radiotherapy model, it was observed that the ratio of FL_Ex980,1550_/FL_Ex808,1550_ increased significantly with cumulative X-ray doses (from 64 on Day 0 to 80 on day 3 and over 100 on day 6), providing real-time feedback on treatment response and supporting timely optimization of radiotherapy regimens (Figure 41b,c). These results showed that this probe had great potential to quantitatively visualize caspase-3 activity in vivo.

It should be noted that in a NRET effect-based internal reference probe, the NRET system may form upon response to the target stimulus. For example, Yang et al. developed a NIR-II ratiometric probe (DCNP@DNA2@IR806, **35**) by covalently connecting DCNP and IR806 with miR-21-responsive DNA strand HDNA1-DNA2, enabling real-time, quantitative in vivo monitoring of miR-21 based on NRET (Figure 42a) [211]. Upon hybridization of HDNA1-DNA2 with miR-21, the proximity between IR806 and DCNP in **35** triggered a NRET effect that enhanced the 1550 nm emission of DCNP under 808 nm excitation (F_1550,Ex808)_, while the 1550 nm emission under 980 nm excitation (F_1550,Ex980_) remains essentially constant (Figure 42a). At 18 h after i.v. injection of **35** into 4T1 tumor models, the ratio F_1550,Ex808_/F_1550,Ex980_ increased with tumor volume, enabling real-time tracking of tumor initiation and progression and facilitating early diagnosis (Figure 42b,c).

Internal-reference NIR-II ratiometric probes leverage a “target-sensitive plus target-inert” self-calibration framework to minimize the impact of systemic perturbations—such as probe distribution, excitation drift, and tissue optical heterogeneity—thereby enhancing the credibility of in vivo quantification. Representative systems targeting H_2_S, NO, •OH, miRNA, and radiotherapy-related enzyme activity have demonstrated sensitive and specific quantitative imaging across multiple tumor models, enabling therapeutic assessment and precision intervention.

Future research on internal-reference NIR-II ratiometric probes should focus on: (1) Spectral and instrument orthogonality—optimizing channel separation and device-compatible designs, such as dual-excitation or single-excitation dual-emission systems; (2) colocalization and PK matching—designing materials to ensure co-localized emitters and consistent pharmacokinetics through unified carriers and interfacial control; (3) standardized workflows—establishing consistent ratiometric imaging protocols and cross-platform calibration to enhance comparability and translational readiness.

## 5. Summary and Outlook

NIR-II fluorescent probes based on organic small-molecule fluorophores have emerged as a versatile platform for noninvasive, in vivo tumor imaging owing to their favorable optical properties, biocompatibility, and tunable pharmacokinetics. This review covers recent advances in molecular design and optical optimization, such as enhanced quantum yield and red-shifted emission, and their applications in D-A-D scaffolds, cyanines, BODIPY, and xanthene-based dyes. Two principal imaging strategies have matured: (1) “Always-on” probes that accumulate in tumors via the EPR effect or through active targeting (for example, antibodies, peptides); (2) “activatable” probes that respond to TME biomarkers (for example, enzymes, pH, reactive oxygen species) to achieve higher signal-to-background ratios and greater specificity. Collectively, these probes show substantial promise for delineating tumor margins, detecting micrometastases, guiding surgical resection, and enabling theranostics.

However, the reported probes all have advantages and limitations. Always-on probes, particularly those with passive targeting, are easy to prepare and convenient to apply, making them promising for clinical translation. Nevertheless, their imaging contrast is inherently limited by accumulation efficiency and is prone to high background signals. Though active-targeting probes enhance specificity and accumulation at the target site, yielding higher SBRs, their effectiveness may be affected by heterogeneous receptor expression and the possible immunogenicity of targeting ligands in the tumor microenvironment. Activatable probes represent a significant advancement in specificity, as they exhibit low background signals and generate a strong turn-on response to specific biomarkers, thereby enabling the detection of pathological activities. Nonetheless, their design is complex, and their fluorescence performance may be affected by target biomarker variability and concentration in the complex tumor microenvironment. Ratiometric probes possess a self-calibrating function that reduces interference from non-target factors and improves quantification reliability. However, this benefit requires more complex molecular or nanoscale engineering and may complicate data acquisition and interpretation. Therefore, before choosing a probe for biomedical applications, trade-offs must be considered based on the biological question, the required specificity and quantitative demands, as well as the feasibility of clinical translation.

In addition, some key challenges remain to be overcome for the clinical translation of NIR-II fluorescent probes based on organic small-molecule NIR-II fluorophores. (1) The trade-off between brightness and wavelength: Extending conjugation for longer-wavelength emission typically lowers fluorescence quantum yield, making it difficult to optimize both brightness and penetration depth simultaneously. (2) Insufficient in vivo characterization: Metabolic pathways, long-term toxicity, and immunogenicity are not fully understood, and targeted probes need improved molecular specificity. (3) TME heterogeneity: Spatial and temporal variations may lead to inconsistent responses in activatable probes, reducing reliability and generalizability.

To address the aforementioned issues, research efforts should focus on the following aspects. (1) Novel luminophores and AI-assisted design: Move beyond conventional frameworks by exploring new emissive cores and leveraging computational and AI tools to co-optimize emission wavelength and quantum yield. (2) Multimodal integration: Combine NIR-II fluorescence with PA, photothermal, ultrasound, MRI, or CT to provide complementary, multidimensional information and enhance deep-tissue quantification. (3) Logic-gated activatable probes: Implement multi-stimuli (for example, AND/OR) designs that require co-localized TME cues to enhance specificity and reduce false positives. (4) Dynamic theranostic monitoring: Develop probes to monitor treatment responses in real time, such as, apoptosis, immune infiltration, drug resistance, to support adaptive, individualized therapy. (5) Translational standardization: Establish standardized manufacturing and evaluation pipelines, conduct systematic biosafety assessments, and expand into procedural settings such as endoscopy and interventional radiology to bridge the translational gap.

In summary, NIR-II fluorescent probes based on organic small-molecule NIR-II fluorophores offer a powerful avenue for precise tumor imaging. With continued innovation in molecular engineering, intelligent imaging strategies, and standardized translational pathways, these technologies are poised to drive significant advances in early diagnosis, image-guided surgery, and personalized cancer treatment.

## Figures and Tables

**Figure 1 sensors-25-07080-f001:**
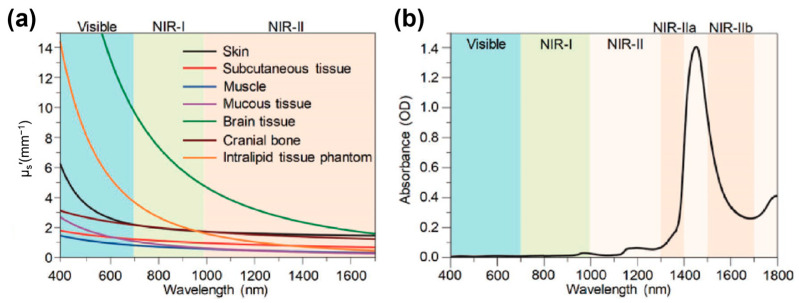
(**a**) Reduced scattering coefficients (μ_s_′) of various tissues plotted as a function of wavelength across 400–1700 nm. (**b**) Visible-near-infrared (Vis–NIR) absorption spectrum of water. OD represents optical density. NIR-IIa: 1300–1400 nm; NIR-IIb: 1500–1700 nm. Reproduced from ref. [45] Copyright (2018), with permission from Wiley-VCH.

**Figure 2 sensors-25-07080-f002:**
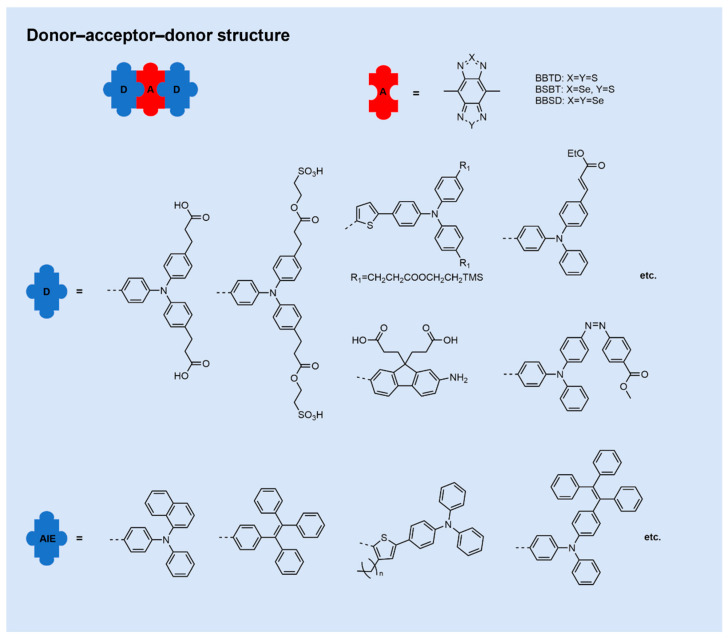
D-A-D structured NIR-II organic small-molecule fluorophores.

**Figure 3 sensors-25-07080-f003:**
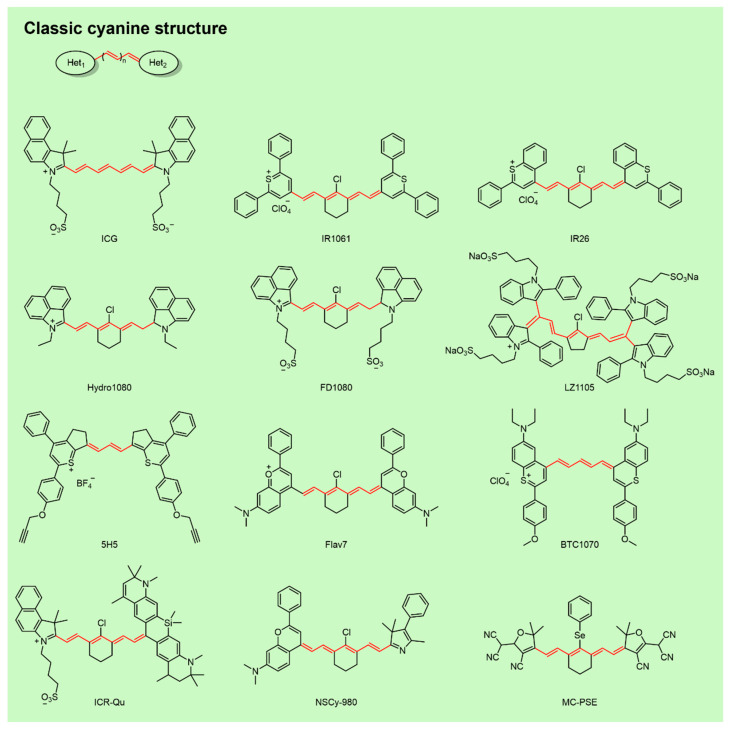
NIR-II cyanine fluorophores.

**Figure 4 sensors-25-07080-f004:**
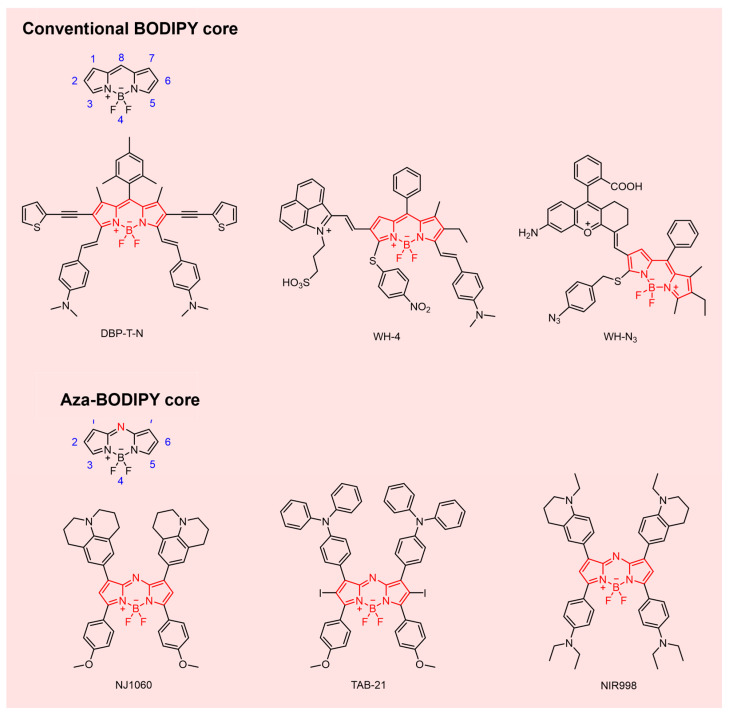
NIR-II BODIPY fluorophores.

**Figure 5 sensors-25-07080-f005:**
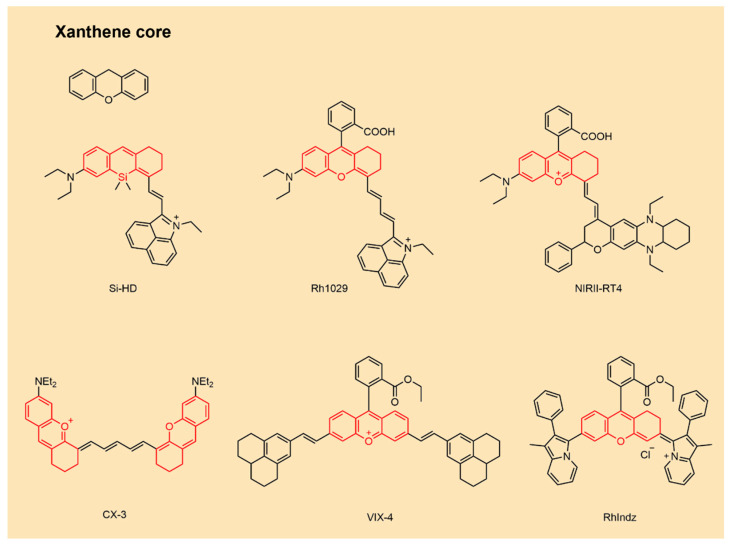
NIR-II Xanthene organic fluorophores.

**Figure 6 sensors-25-07080-f006:**
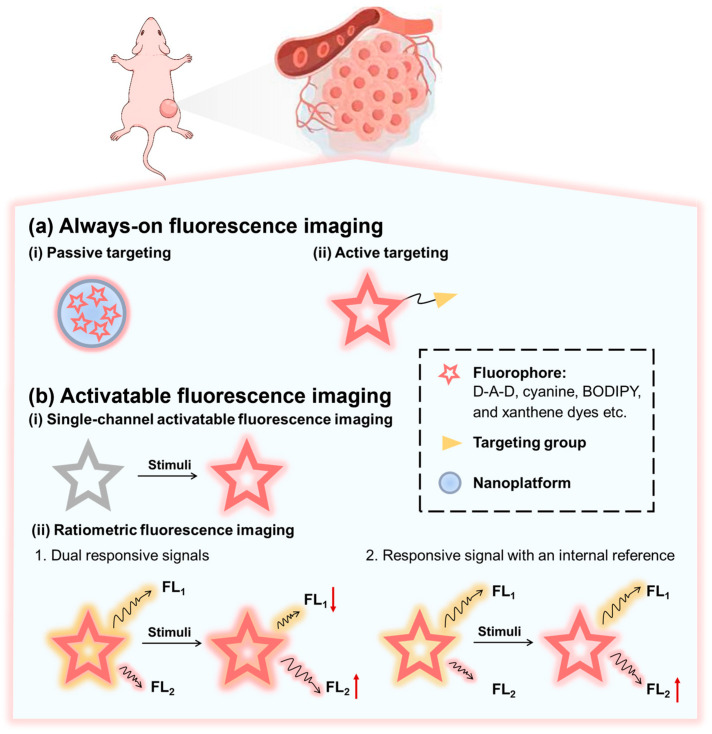
Schematic diagram showing the two design strategies of NIR-II fluorescent probes for tumor imaging using organic small-molecule fluorophores. (**a**) Always-on imaging strategy, including nano-platform-based passive targeting strategy (**i**) and targeting group-mediated active targeting strategy (**ii**). (**b**) Activatable imaging strategy, including single-channel activatable fluorescence imaging (**i**) and ratiometric fluorescence imaging (**ii**). The wavy arrows represent the fluorescence signals at distinct wavelengths, labeled FL_1_ and FL_2_.

**Figure 7 sensors-25-07080-f007:**
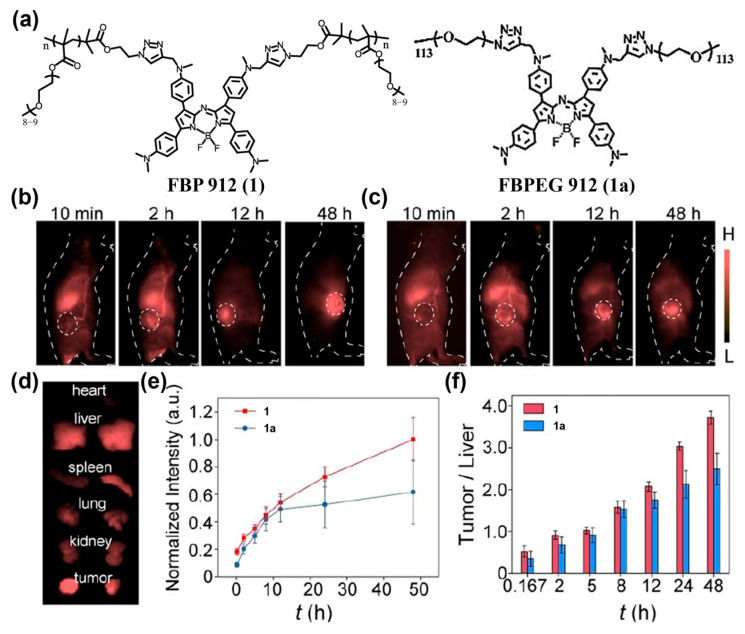
(**a**) The chemical structures of FBP 912 (**1**) and FBPEG 912 (**1a**). (**b**,**c**) In vivo NIR-II fluorescence imaging of subcutaneous tumors using **1** (**b**) or **1a** (**c**). (**d**) Ex vivo NIR-II imaging of main organs and tumors at 48 h post-injection, left: **1**; right: **1a**. (**e**,**f**) Normalized intensity at the tumor region (**e**) and tumor/liver ratio (**f**) at different time point after i.v. injection of **1** or **1a** into mice. Reproduced from ref. [138] Copyright (2021), with permission from Wiley-VCH.

**Figure 8 sensors-25-07080-f008:**
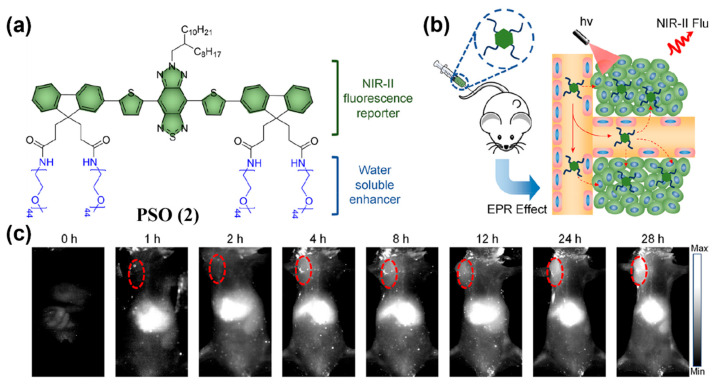
(**a**) Chemical structure of PSO (**2**). (**b**) Schematic illustration showing the use of **2** for NIR-II fluorescence imaging of tumor. (**c**) Images of **2**-injected tumor-bearing mice at different time points captured by the NIR-II fluorescence imaging system. The red circles in the images indicate the tumor location. Reproduced from ref. [139] Copyright (2022), with permission from American Chemical Society.

**Figure 9 sensors-25-07080-f009:**
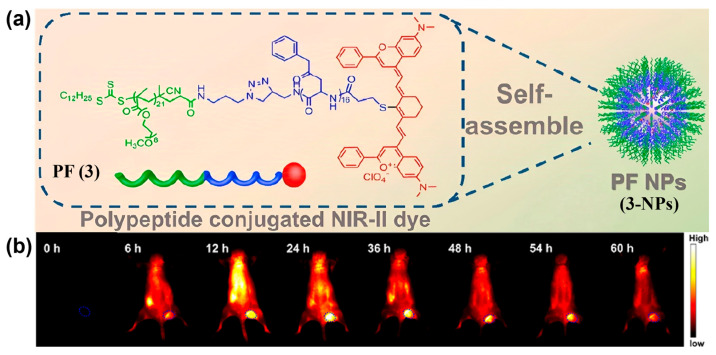
(**a**) Schematic diagram showing the self-assembly of PF (**3**) into PF-NPs (**3-NPs**). (**b**) Representative in vivo NIR-II fluorescence imaging of tumor-bearing mice at different time points after i.v. injection of **3-NPs**. Reproduced from ref. [142] Copyright (2019), with permission from American Chemical Society.

**Figure 10 sensors-25-07080-f010:**
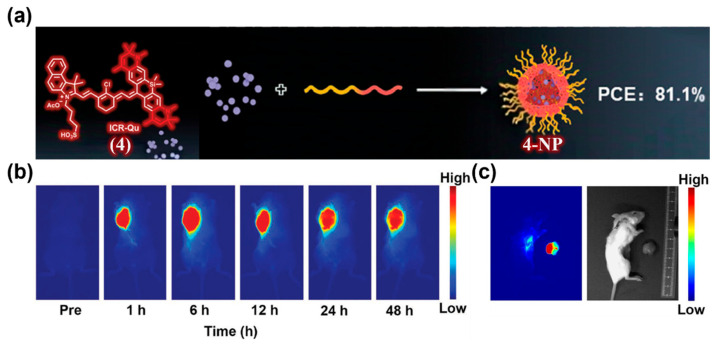
(**a**) Schematic diagram showing preparation of **4-NP** by encapsulating ICR-Qu (**4**) within DSPE-PEG_2000_. (**b**) NIR-II fluorescence imaging (using an 808 nm excitation laser) of tumor at various time points (0, 1, 6, 12, 24, and 48 h) after administration of **4-NP** into 4T1 tumor-bearing BALB/c mice. (**c**) NIR-II fluorescence imaging-guided tumor resection (using an 808 nm excitation laser). Reproduced from ref. [146] Copyright (2023), with permission from Wiley-VCH.

**Figure 11 sensors-25-07080-f011:**
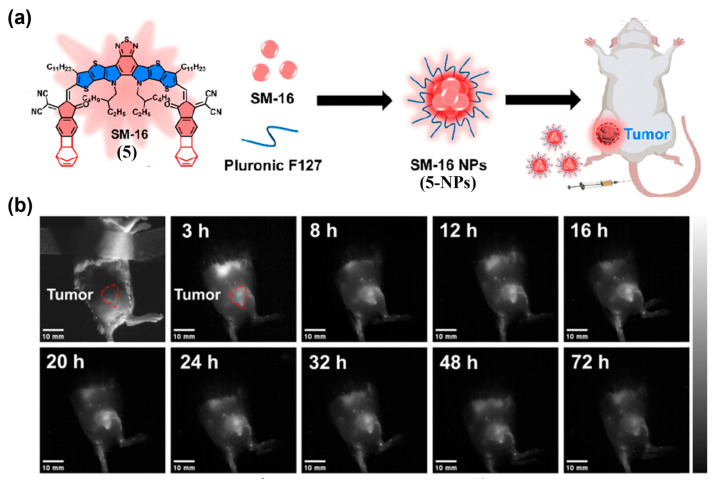
(**a**) Structural modification and assembly process of SM-16 (**5**) with Pluronic F127 to form SM-16 NPs (**5-NPs**). (**b**) Time-dependent NIR-II fluorescence imaging of 4T1-tumor-bearing mice i.v. injected with **5-NPs**. Reproduced from ref. [148] Copyright (2024), with permission from Royal Society of Chemistry.

**Figure 12 sensors-25-07080-f012:**
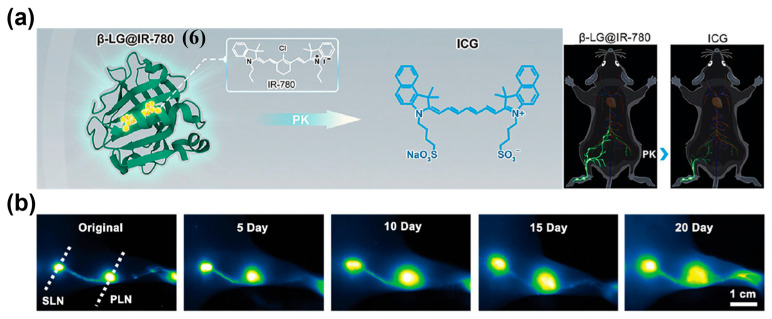
(**a**) NIR-II lymphography with ICG or *β*-LG@IR-780 (**6**) probes. (**b**) NIR-II popliteal and sacral lymph nodes (PLN and SLN) imaging at different time points after tumor inoculation using **6**. Reproduced from ref. [151] Copyright (2023), with permission from Wiley-VCH.

**Figure 13 sensors-25-07080-f013:**
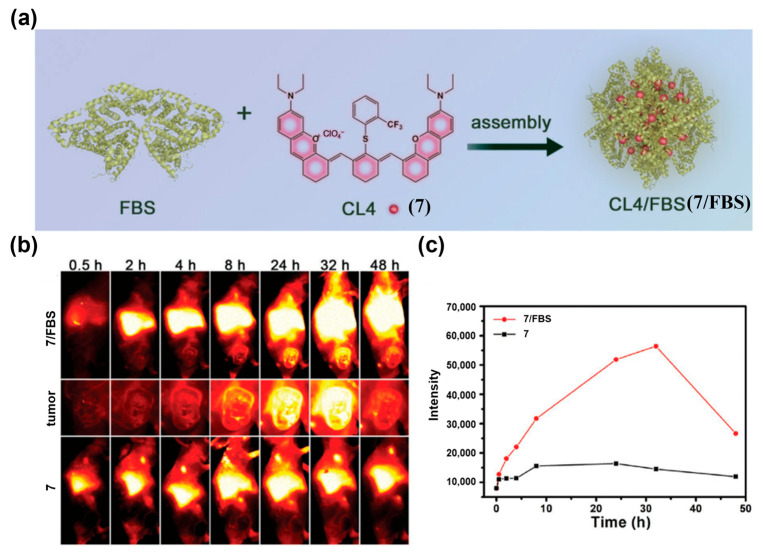
(**a**) Schematic diagram of the construction of CL4/FBS (**7/FBS**). (**b**)Time-course whole-body NIR-II fluorescence images of 4T1 tumor-bearing mice and magnified images of the tumor after i.v. injection of **7/FBS** or **7**. (**c**) Quantified NIR-II fluorescence intensities of tumors in (**b**). Reproduced from ref. [152] Copyright (2022), with permission from Wiley-VCH.

**Figure 14 sensors-25-07080-f014:**
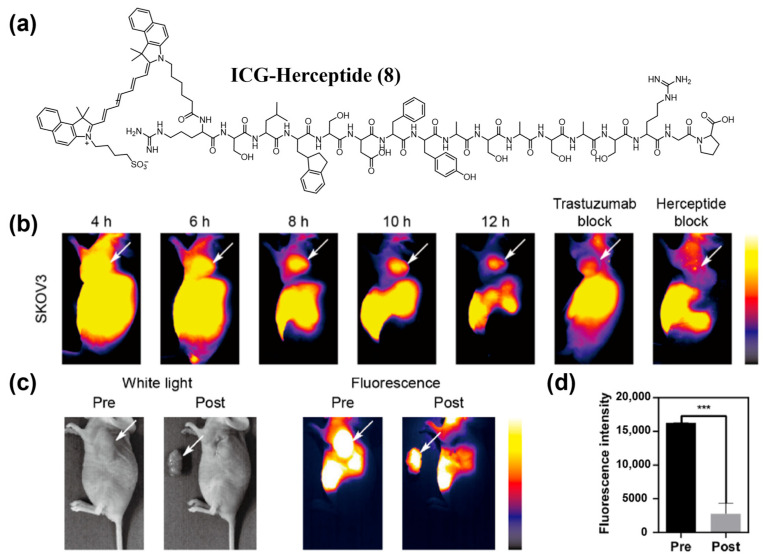
(**a**) Chemical structure of ICG-Herceptide probe (**8**). (**b**) NIR-II fluorescence imaging of tumor-bearing mice at 0−12 h after i.v. injection of **8** with or without the injection of trastuzumab and Herceptide. (**c**) NIR-II fluorescence imaging-guided surgery using **8**. (**d**) Fluorescence intensity analysis of images in (**c**). *** *p* < 0.001. Reproduced from ref. [153] Copyright (2023), with permission from American Chemical Society.

**Figure 15 sensors-25-07080-f015:**
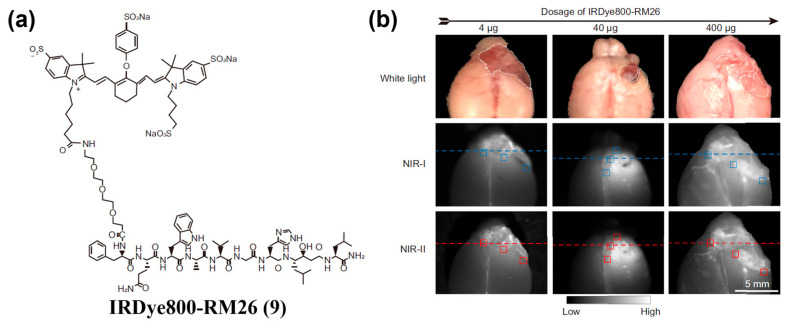
(**a**) Molecular structure of IRDye800-RM26 (**9**). (**b**) Ex vivo NIR-I/II imaging and cross-sectional intensity profile (blue dashed bars in images under NIR-I window, or red-dashed bars in images under NIR-II window) of glioblastoma (white dotted lines) using different dosages of **9** at NIR-I and NIR-II windows. Reproduced from ref. [154] Copyright (2023), with permission from Wiley-VCH.

**Figure 16 sensors-25-07080-f016:**
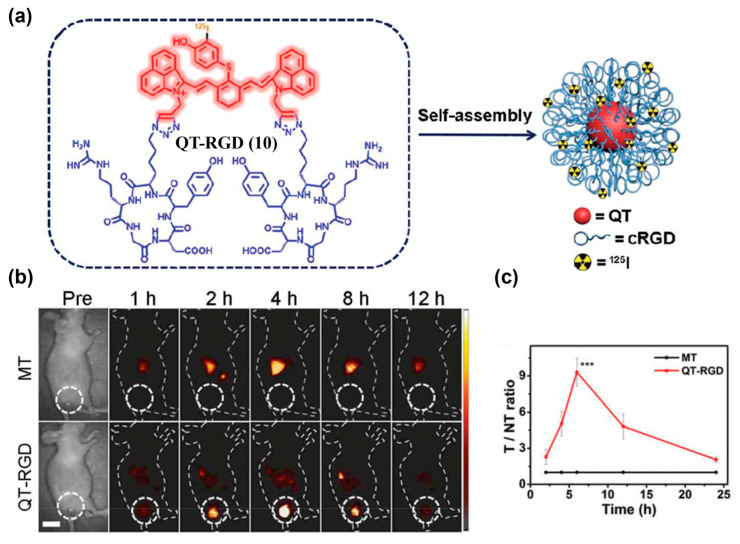
(**a**) Schematic diagram showing the self-assembly of QT-RGD (**10**) into nanoparticles. (**b**) In vivo NIR-II fluorescence images of 4T1 tumor-bearing mice at various time points (0, 1, 2, 4, 8 and 12 h) after i.v. injection of MT and **10**. (**c**) Quantitative analysis of the fluorescence intensities of the tumor sites at different time points in (**b**). *** *p* < 0.001. Reproduced from ref. [155] Copyright (2020), with permission from Royal Society of Chemistry.

**Figure 17 sensors-25-07080-f017:**
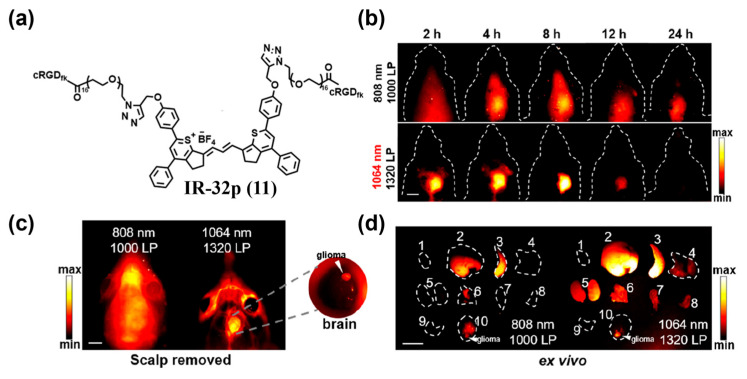
(**a**) Chemical structure of probe IR-32p (**11**). (**b**) NIR-II and NIR-IIa fluorescence imaging of orthotopic glioma mice after i.v. injection of **11**. (**c**) NIR-II/IIa glioma imaging (scalp removed) and ex vivo brain at 24 h p.i. (**d**) Ex vivo NIR-II and NIR-IIa imaging results of the main organs. Nos. 1–10: heart, liver, spleen, lung, kidney, stomach, intestine, muscle, skin, and brain, respectively. Organs were extracted at 24 h post injection. NIR-II fluorescence images were captured under 808 nm laser irradiation (300 mW cm^−2^, 1000 LP) at 100 ms (**b**,**c**) and 300 ms (**d**) exposure time. NIR-IIa images were captured under 1064 nm laser irradiation (500 mW cm^−2^, 1320 nm LP, 400 ms). Reproduced from ref. [156] Copyright (2024), with permission from American Chemical Society.

**Figure 18 sensors-25-07080-f018:**
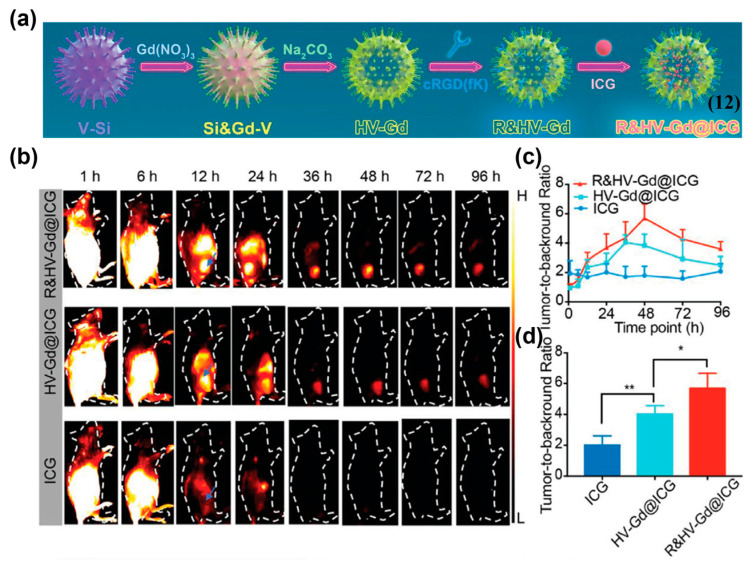
(**a**) Schematic illustrating the fabrication of R&HV-Gd@ICG (**12**). (**b**) NIR-II fluorescence images of 4T1-tumor-bearing mice intravenously injected with **12**, HV-Gd@ICG, or ICG, respectively. (**c**) TBR concentration plotted as function of time for 4T1-tumor-bearing mice (*n* = 4). (**d**) Comparison of individual **12**, HV-Gd@ICG, and ICG specific tumor-targeting capabilities at maximum TBR (*n* = 4). * *p* < 0.05, ** *p* < 0.01. Reproduced from ref. [157] Copyright (2022), with permission from Wiley-VCH.

**Figure 19 sensors-25-07080-f019:**
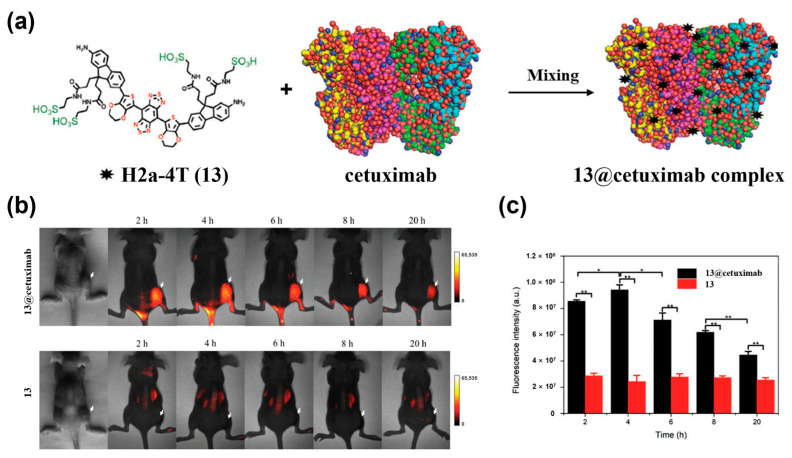
(**a**) Schematic illustration showing the formation of H2a-4T@Cetuximab complex (**13@cetuximab** complex). (**b**) Representative NIR-II fluorescence images (1000 LP and 100 ms) of the HCT116 tumor model at 2, 4, 6, 8, and 20 h after tail vein injection of the **13**@Cetuximab complex and **13** under an 808 nm excitation. (**c**) Quantification analysis of fluorescence signals of the tumor regions in (**b**) (*n* = 3). * *p* < 0.05, ** *p* < 0.01. Reproduced from ref. [69] Copyright (2018), with permission from Wiley-VCH.

**Figure 20 sensors-25-07080-f020:**
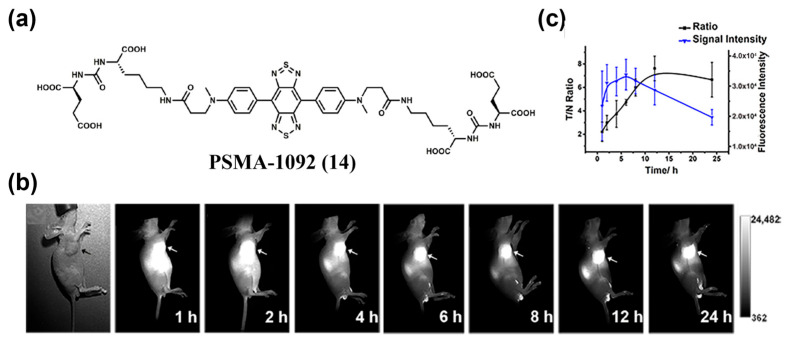
(**a**) The chemical structure of PSM-1092 (**14**). (**b**) Time-course NIR-II fluorescence images of LNCaP tumor-bearing mice after tail vein injection. (**c**) TNR (left Y axis, black line) and tumor uptake (right axis, blue line) versus time curve after intravenous injection of **14** in tumor-bearing mice. Reproduced from ref. [167] Copyright (2021), with permission from American Chemical Society.

**Figure 21 sensors-25-07080-f021:**
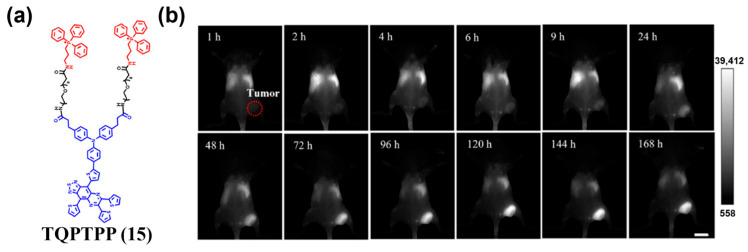
(**a**) The chemical structure of TQPTPP (**15**). (**b**) Time-course NIR-II fluorescence imaging (808 nm excitation, 1000 long-pass filter, 100 ms exposure) of tumors with tail vein injection of **15**. Reproduced from ref. [168] Copyright (2023), with permission from Wiley-VCH.

**Figure 22 sensors-25-07080-f022:**
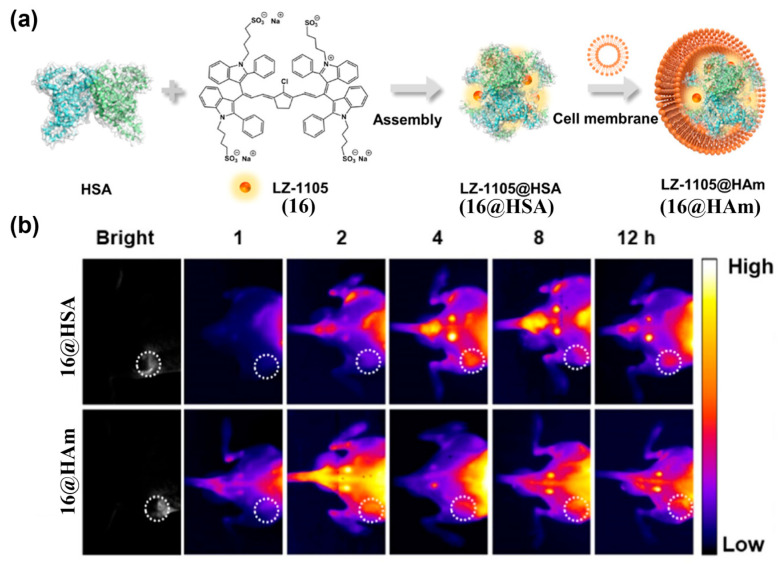
(**a**) Schematic illustration of homologous cell membrane-coated albumin-energized NIR-II Cyanine Dye LZ-1105@ Ham (**16@HAm**). (**b**) Representative NIR-II fluorescence images of A549 tumor-bearing mice at various time points after i.v. injection of **16@HSA** and **16@HAm**. Reproduced from ref. [172] Copyright (2025), with permission from American Chemical Society.

**Figure 23 sensors-25-07080-f023:**
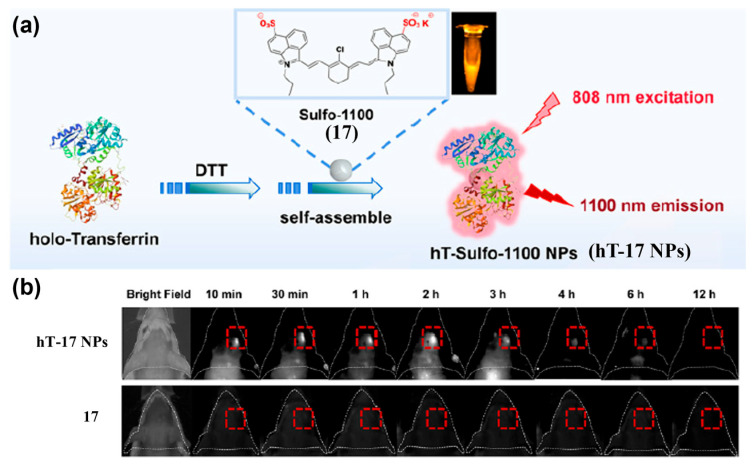
(**a**) Schematic representation of the basic steps for fabrication of hT-Sulfo-1100 NPs (**hT-17 NPs**). (**b**) NIR-II fluorescence images of orthotopic glioma. The red dotted boxes mark the locations of tumors. Reproduced from ref. [173] Copyright (2024), with permission from Royal Society of Chemistry.

**Figure 24 sensors-25-07080-f024:**
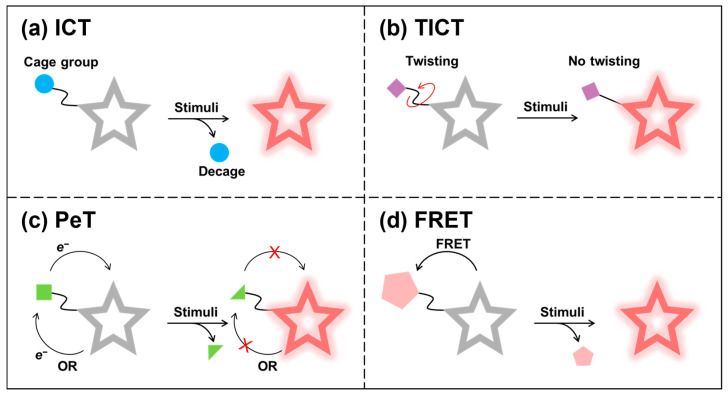
Cartoon illustration of different mechanisms to design stimuli-activated fluorescent probes, including: (**a**) recovery of ICT effect, (**b**) restriction of TICT, (**c**) inhibition of PeT effect, and (**d**) elimination of FRET.

**Figure 25 sensors-25-07080-f025:**
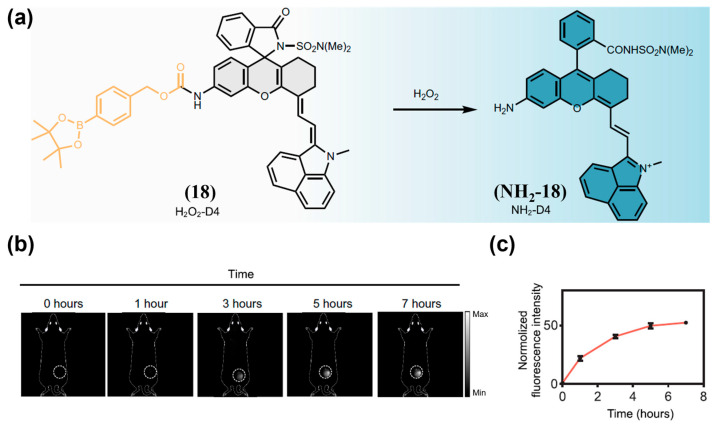
(**a**) Chemical structure of H_2_O_2_-D4 (**18**) and the sensing mechanism towards H_2_O_2_. (**b**) NIR-II fluorescence images of 4T1 tumor-bearing mice before (0 h) and after intratumor treatment with **18** for 1, 3, 5, and 7 h. (**c**) Normalized fluorescence intensity of the tumor region in (**b)**. Reproduced from ref. [190] Copyright (2024), with permission from AAAS.

**Figure 26 sensors-25-07080-f026:**
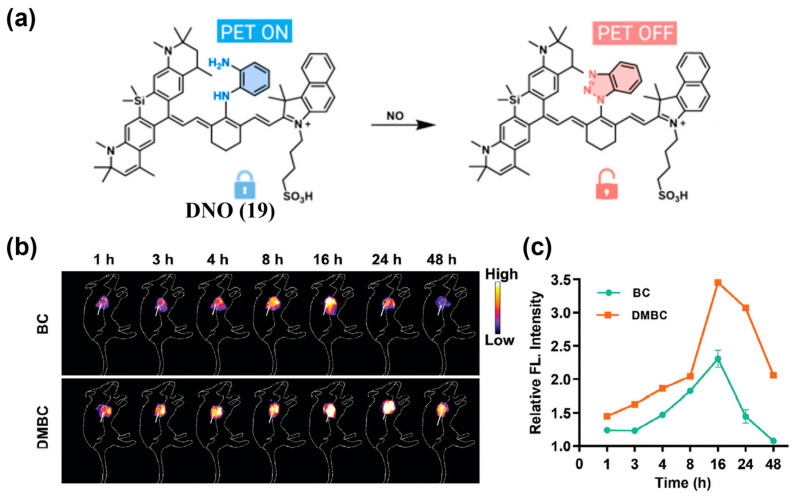
(**a**) Chemical structure of NO-responsive molecule DNO (**19**) and the response mechanism towards NO. (**b**) Time-course NIR-II fluorescence images of BALB/c mice with subcutaneous breast tumor (BC group) and diabetic BALB/c mice with subcutaneous breast tumor (DMBC group) after injecting DNPS under 808 nm laser irradiation. (**c**) Quantified data from panel (**b**). Reproduced from ref. [191] Copyright (2025), with permission from Wiley-VCH.

**Figure 27 sensors-25-07080-f027:**
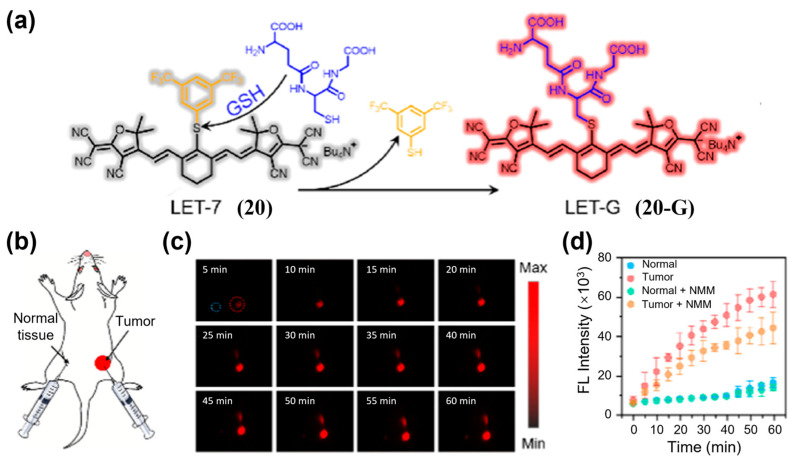
(**a**) Illustration of a GSH-activatable NIR-II fluorescent probe LET-7 (**20**) for visualization of GSH. (**b**) Schematic diagram of probe injection in tumor-bearing mice. (**c**) Time-dependent NIR-II fluorescence images of 4T1 tumor-bearing mice after intratumoral (tumor side, red cycle) and subcutaneously (normal tissue side, blue cycle) injection of **20**, respectively. (**d**) The corresponding quantitative NIR-II fluorescence intensity of tumor and normal tissues. Reproduced from ref. [192] Copyright (2021), with permission from American Chemical Society.

**Figure 28 sensors-25-07080-f028:**
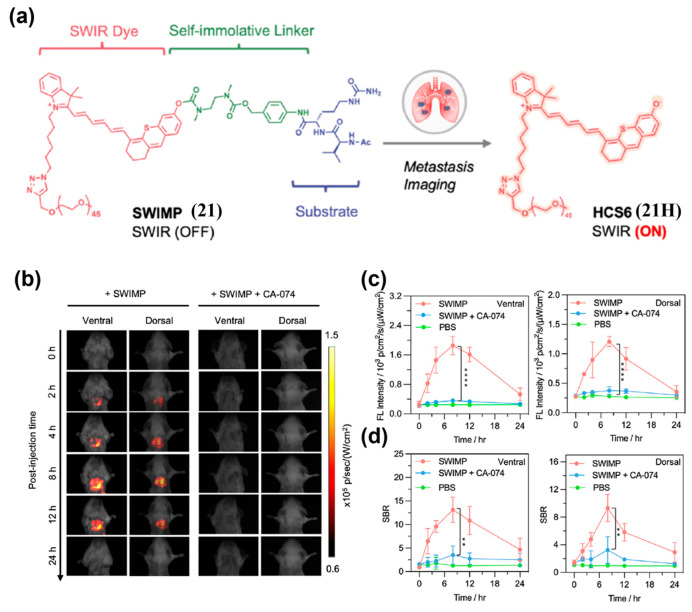
(**a**) Chemical structures of SWIMP (**21**) and response mechanism of **21** toward Cath B for lung metastasis imaging. (**b**) Representative in vivo SWIR fluorescence images of living mice at different time points after intratracheal injection of **21** (left panel) with or without pretreatment by inhibitor (right panel). Excitation: 808 nm; long-pass filter: 950 nm; acquisition time: 1 s. (**c**) Quantification and (**d**) SBRs of SWIR fluorescence in the tumor region as a function of time from both the ventral (left) and dorsal (right) sides of the mice (*n* = 3). ** *p* < 0.01, **** *p* < 0.001. Reproduced from ref. [193] Copyright (2025), with permission from American Chemical Society.

**Figure 29 sensors-25-07080-f029:**
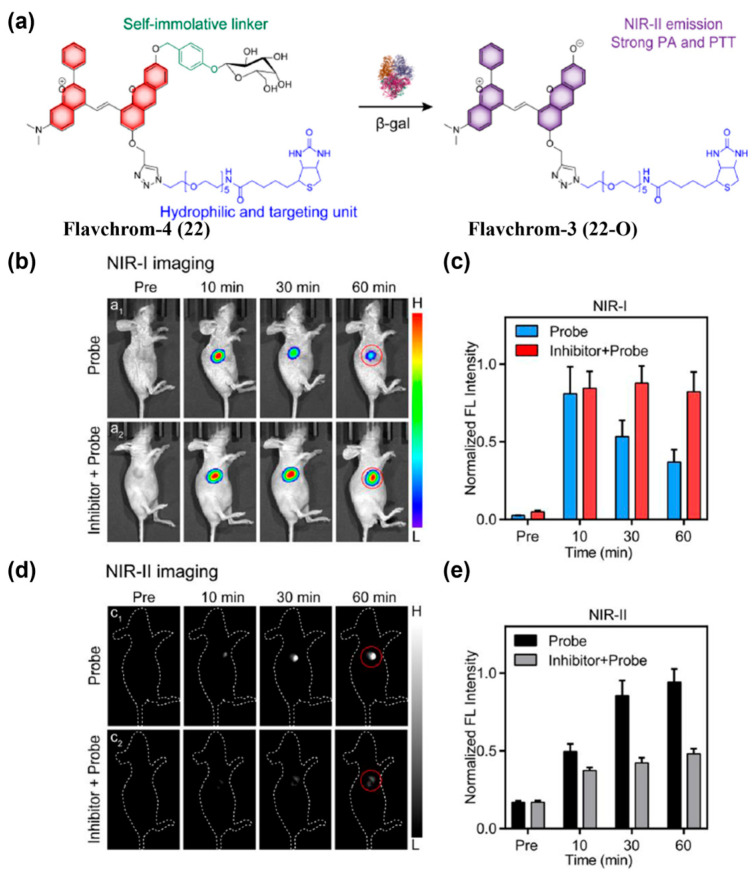
(**a**) Design of the *β*-gal-activated fluorescent probe, Flavchrom 4 (**22**). (**b**,**c**) Time-dependent NIR-I imaging (**b**) and normalized fluorescence signal values at tumor site (**c**) of SKOV3 tumor-bearing mouse models after intratumor injection with **22.** (**d**,**e**) Time-dependent NIR-II imaging (**d**) and normalized fluorescence signal values at tumor site (**e**) of SKOV3 tumor-bearing mouse models after intratumor injection with **22**. Note: red circle indicates the region of the tumor. Reproduced from ref. [194] Copyright (2022), with permission from American Chemical Society.

**Figure 30 sensors-25-07080-f030:**
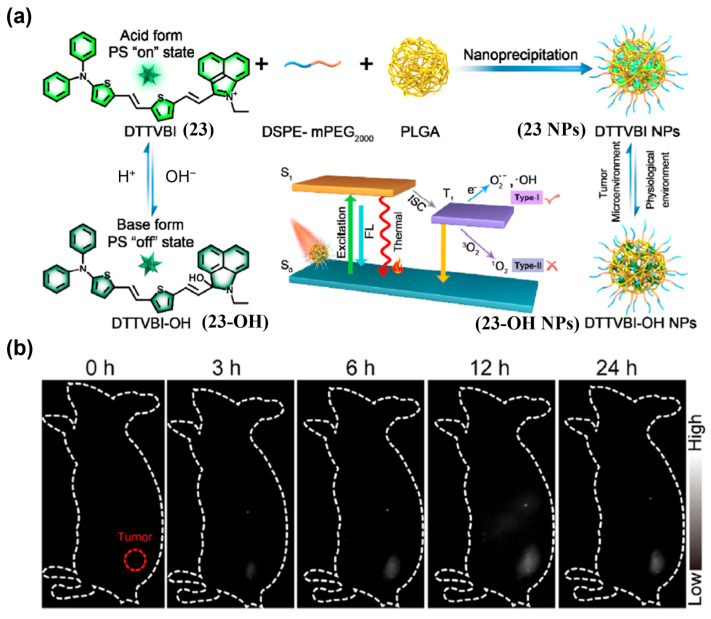
(**a**) Schematic illustration of the response mechanisms of DTTVBI (**23**) and **23 NPs** to pH. (**b**) NIR-II fluorescence imaging of PDX tumor-bearing mice at different time points after intravenous injection of **23 NPs**. Reproduced from ref. [195] Copyright (2023), with permission from American Chemical Society.

**Figure 31 sensors-25-07080-f031:**
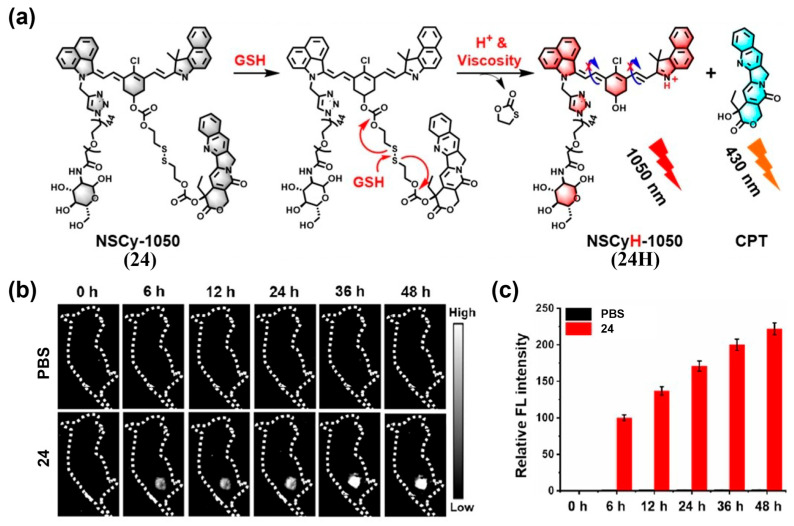
(**a**) The turn-on mechanism of the probe NSCy1050 (**24**) by H^+^ and high viscosity. (**b**) Time-dependent NIR-II fluorescence imaging of 4T1 tumor-bearing mice after i. v. injection of 200 μL PBS or 200 μL **24** (1 mM). (**c**) Relative fluorescence intensities on the subcutaneous 4T1 tumors in (**b**). Reproduced from ref. [197] Copyright (2023), with permission from Wiley-VCH.

**Figure 32 sensors-25-07080-f032:**
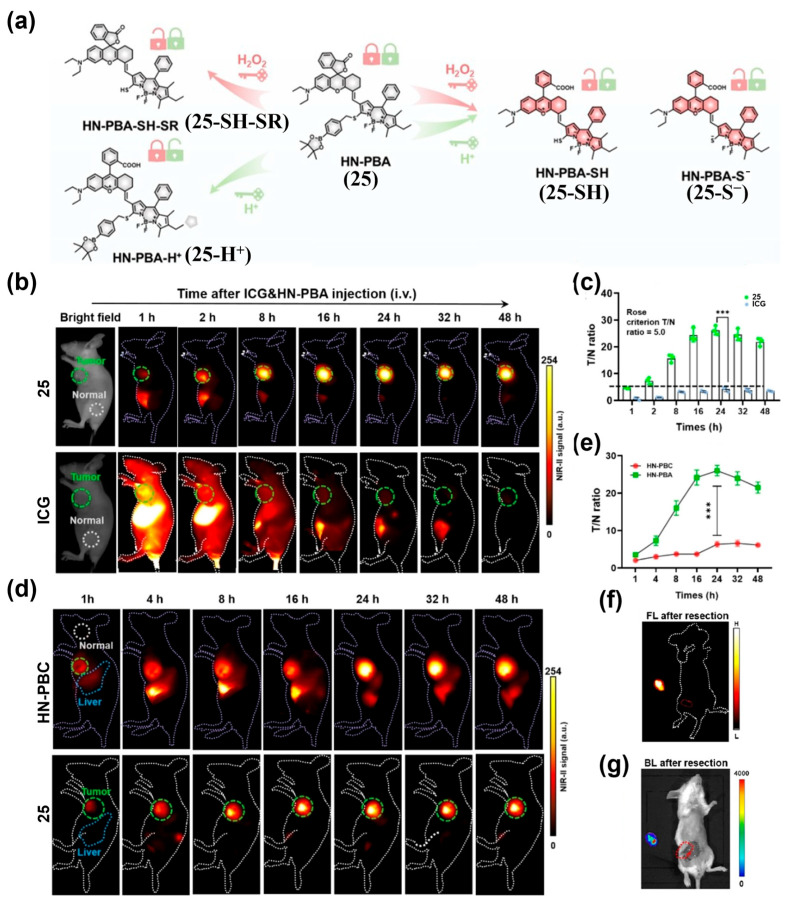
(**a**) Chemical structures of HN-PBA (**25**), HN-PBA-H^+^ (**25-H^+^**), HN-PBA-SH (**25-SH**), HN-PBA-S^−^ (**25-S^−^**), and HN-PBA-SH-SR (**25-SH-SR**) and proposed dual-lock control and response strategy. (**b**) Longitudinal in vivo NIR-II fluorescence images of 4T1 tumor-bearing mice after intravenous injection (i.v.) of **25** or ICG, respectively. (**c**) T/N ratios of **25** and ICG over time. (**d**) Real-time NIR-II imaging of tumor-bearing mice after i.v. injection of **25** and the H_2_O_2_-activated control probe HN-PBC. (**e**) T/N ratios of **25** and HN-PBC over time. (**f**,**g**) NIR-II fluorescence (**f**) and bioluminescence (**g**) images of the tumor resected mice. *** *p* < 0.001. Reproduced from ref. [198] Copyright (2025), with permission from Wiley-VCH.

**Figure 33 sensors-25-07080-f033:**
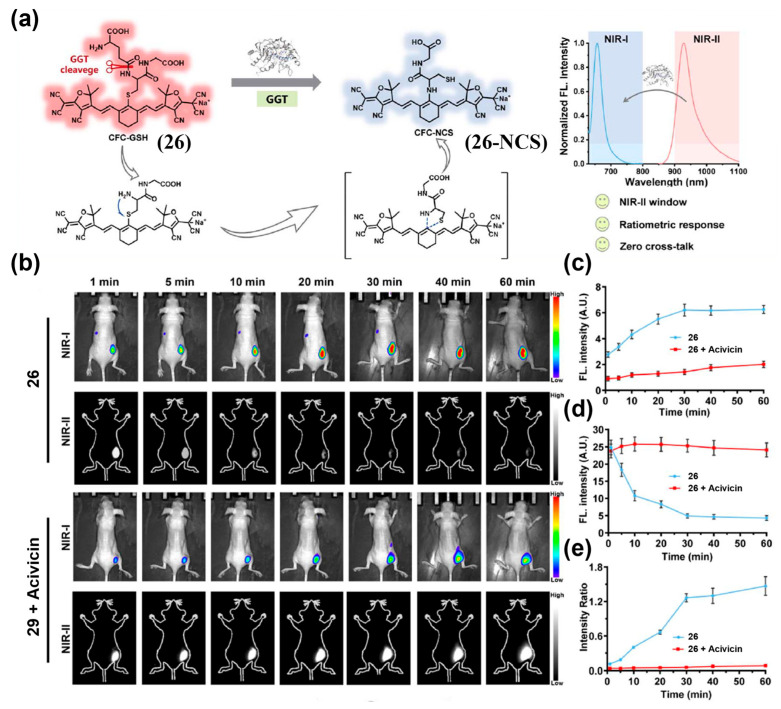
(**a**) Schematic illustration of the sensing mechanism of CFC-GSH (**26**) for GGT. (**b**) NIR-I (λex = 605 nm, λem = 660 nm) and NIR-II (λex = 808 nm, 1075 nm LP) fluorescence imaging of tumor-bearing BALB/c nude mice at different time after intratumoural injection of **26** without/with pretreatment of acivicin for another 30 min. (**c**–**e**) Time-dependent quantitative statistics of NIR-I (**c**) and NIR-II (**d**) fluorescence intensity, and (**e**) the corresponding fluorescence intensity ratio of the mice in (**b**). Reproduced from ref. [202] Copyright (2025), with permission from Royal Society of Chemistry.

**Figure 34 sensors-25-07080-f034:**
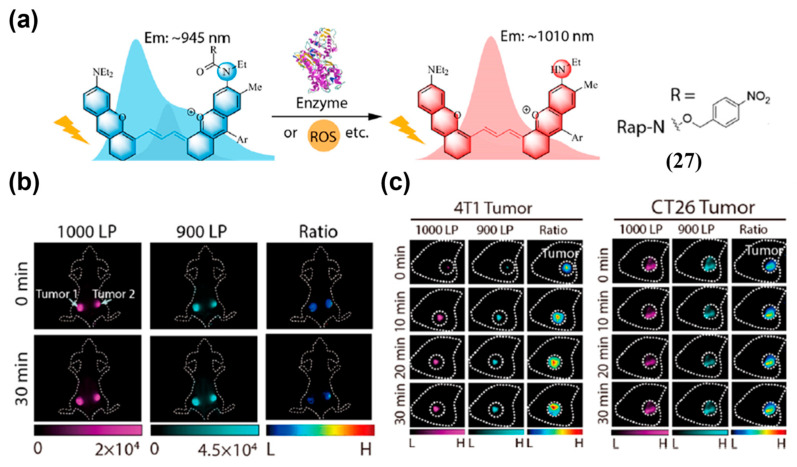
(**a**) Biosensing mechanism of the NIR-II ratiometric probe Rap-N (**27**) toward NTR. (**b**) Time-course NIR-II fluorescence images and corresponding ratiometric images of mice bearing 4T1 tumors, the left tumor was treated with inhibitor dicoumarol. (**c**) Time-course NIR-II fluorescence images of 4T1 and CT-26 tumor. Reproduced from ref. [131] Copyright (2022), with permission from Wiley-VCH.

**Figure 35 sensors-25-07080-f035:**
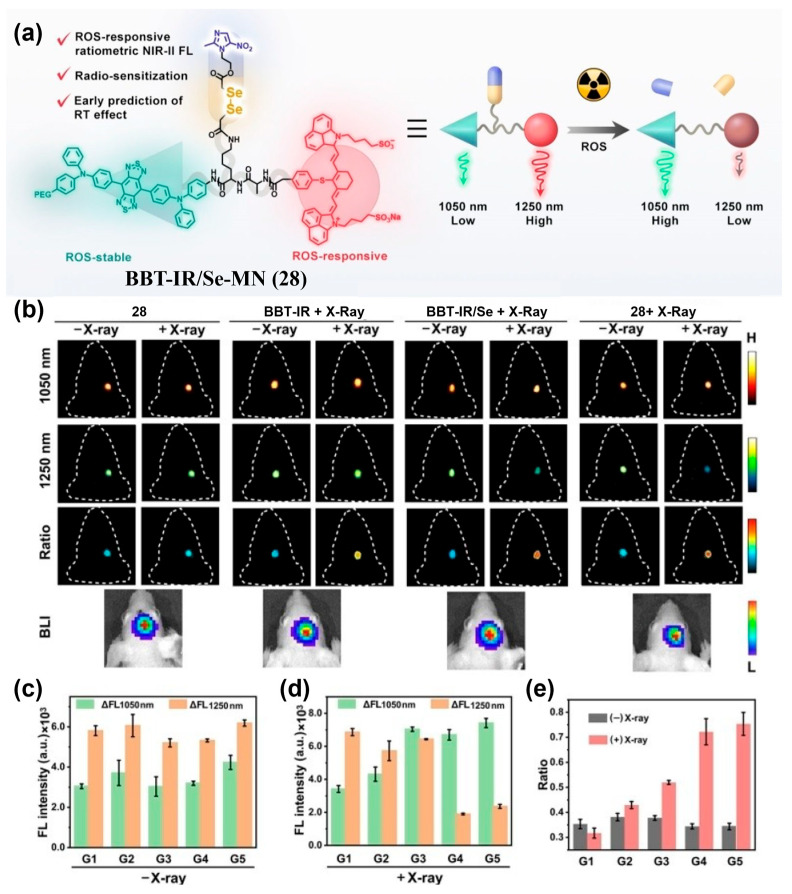
(**a**) Schematic illustration of ROS-responsive NIR-II ratiometric probe BBT-IR/Se-MN (**28**). (**b**) Representative NIR-II fluorescence, ratiometric fluorescence, and bioluminescence images of U87 orthotopic glioma-bearing mice under different treatments. (**c**–**e**) Corresponding NIR-II fluorescence signal intensity of tumors in (**b**). Reproduced from ref. [203] Copyright (2023), with permission from Wiley-VCH.

**Figure 36 sensors-25-07080-f036:**
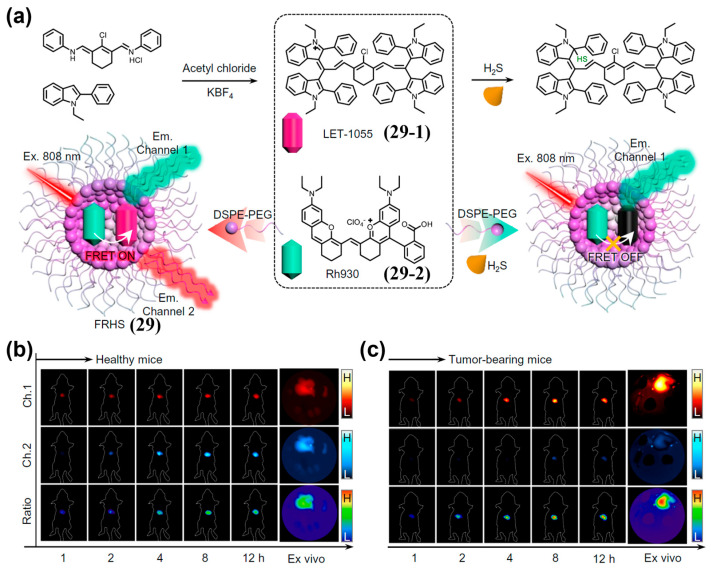
(**a**) Schematic illustration of the construction of the FRET-based ratiometric NIR-II window H_2_S sensor, FRHS (**29**). (**b**,**c**) Time-dependent NIR-II fluorescence images at Ch.1 and Ch.2, and ratiometric Ch.1/Ch.2 fluorescence images of healthy mice (**b**) and tumor-bearing mice (**c**), and corresponding ex vivo organs after i.v. injection of **29**. Reproduced from ref. [204] Copyright (2023), with permission from AAAS.

**Figure 37 sensors-25-07080-f037:**
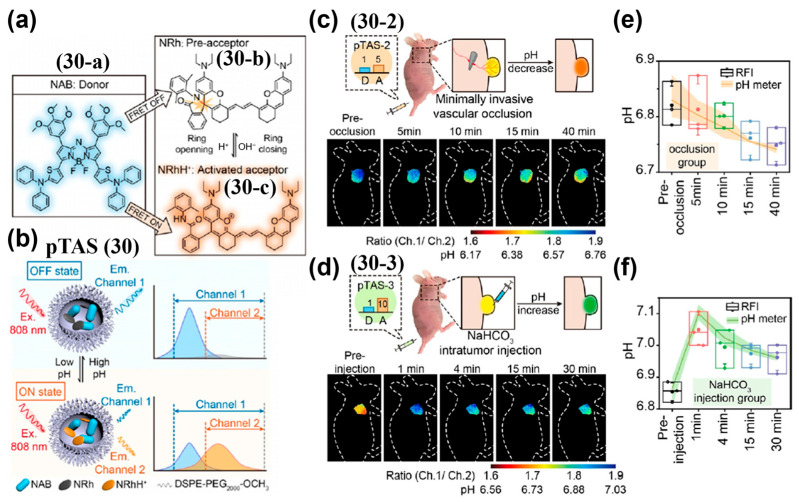
(**a**) Chemical structures of donor NAB (**30-a**), pre-acceptor NRh (**30-b**) and pH activated acceptor NRhH^+^ (**30-c**). (**b**) pH response illustration of pTAS (**30**). Channel 1–2 represent for the signal collection regions during ratiometric fluorescence imaging. (**c**,**d**) Illustrated animal models and ratiometric fluorescence images of tumors during the two pH changing processes after administration of pTAS-2 (**30-2**) and pTAS-3 (**30-3**), respectively. (**e**,**f**) Comparison of the pH changing in two physical processes by ratiometric fluorescence imaging and microelectrode pH meter. Reproduced from ref. [205] Copyright (2021), with permission from Wiley-VCH.

**Figure 38 sensors-25-07080-f038:**
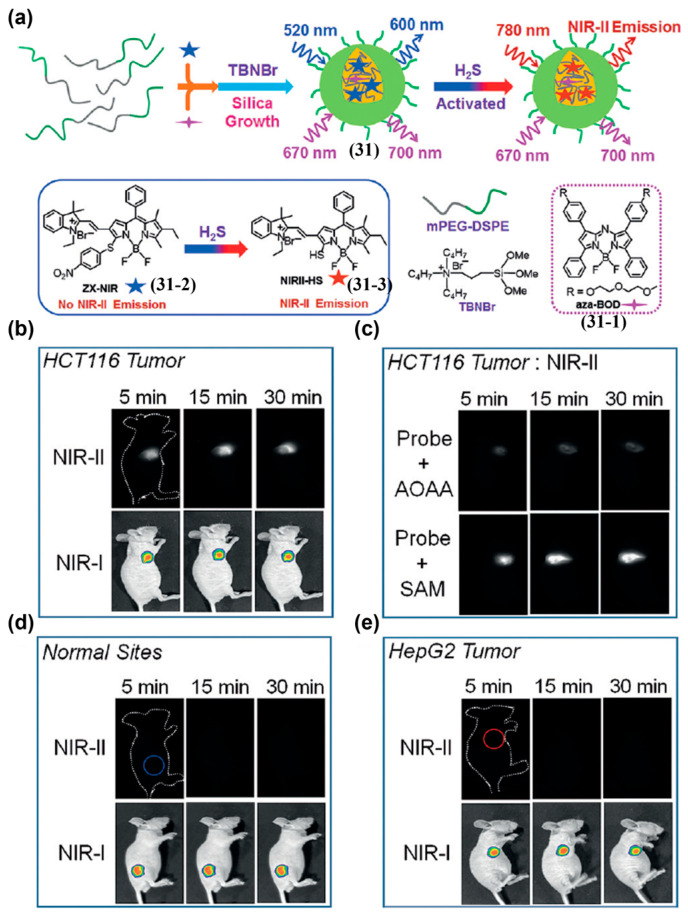
(**a**) Schematic diagram showing the construction of multi-wavelength nanoprobes **31** with activatable emission in the NIR-II window. (**b**–**e**) Visualization and differentiation of cancers based on H_2_S activation of **31**. (**b**) HCT116 tumor-bearing mice. (**c**) The effects of inhibitor (AOAA) and activator (SAM) in HCT116 tumor-bearing mice. (**d**) Normal sites. (**e**) HepG2 tumor-bearing mice. Reproduced from ref. [207] Copyright (2018), with permission from Wiley-VCH.

**Figure 39 sensors-25-07080-f039:**
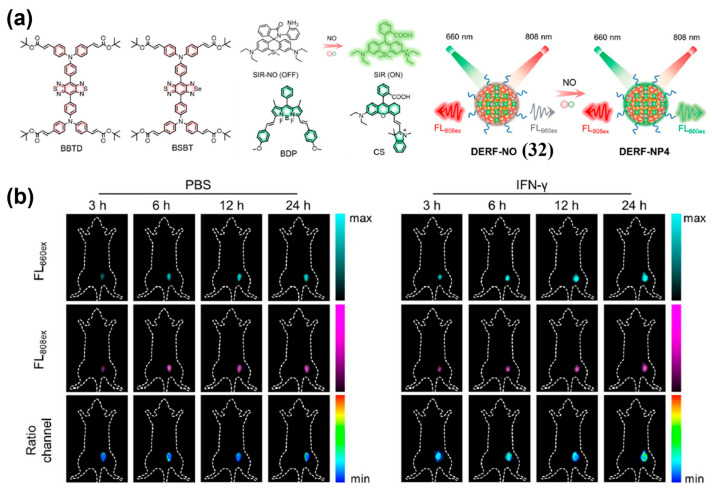
(**a**) Chemical structures of energy acceptors (BBTD and BSBT) and donors (SIR, BDP, and CS), the chemical structure change in SIR-NO before and after reaction with NO, and schematic illustration showing the ratiometric response of DERF-NO (**32**) to NO with the dual-excitation ratiometric detection manner. (**b**) Representative fluorescence images of 4T1 tumor-bearing mice treated with PBS or IFN-*γ* (i.t.) and then i.v.-injected with **32**. Reproduced from ref. [208] Copyright (2025), with permission from Royal Society of Chemistry.

**Figure 40 sensors-25-07080-f040:**
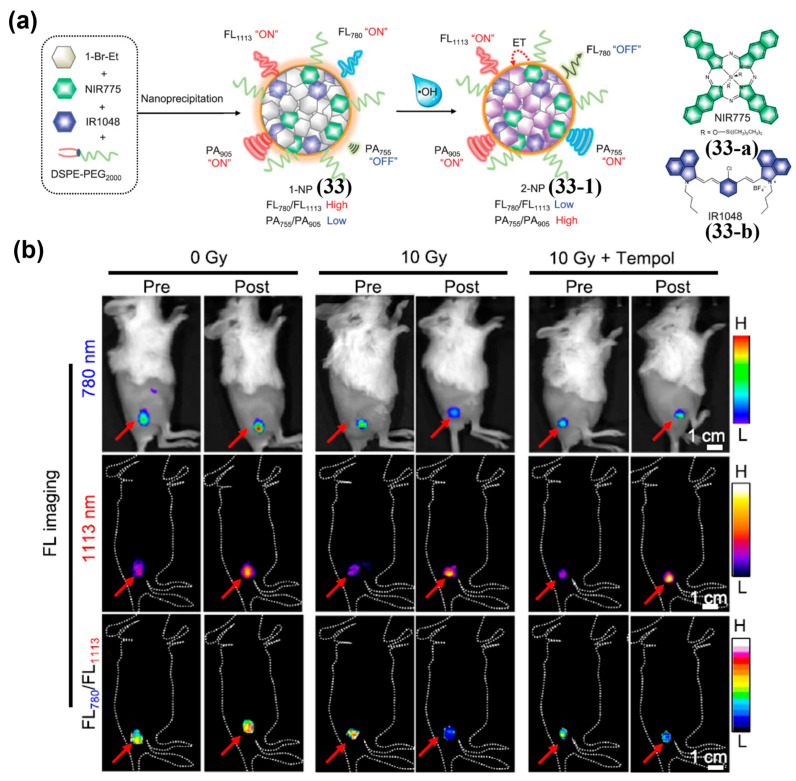
(**a**) Schematic illustration of preparation of 1-NP (**33**) via DSPE-PEG_2000_-assisted encapsulation of 1-BrEt, NIR775 and IR1048, and proposed conversion of **33** into 2-NP (**33-1**) in response to •OH, accompanying by a reduced fluorescence (FL_780_/FL_1113_) ratio but a concurrently increased photoacoustic (PA_755_/PA_905_) ratio. Right: chemical structures of NIR775 and IR1048 used for the preparation of **33**. (**b**) Fluorescence and corresponding ratiometric fluorescence images of 4T1 tumors following indicated treatment. Red arrows and circles indicate the tumor locations. Reproduced from ref. [209] Copyright (2021), with permission from Springer Nature.

**Figure 41 sensors-25-07080-f041:**
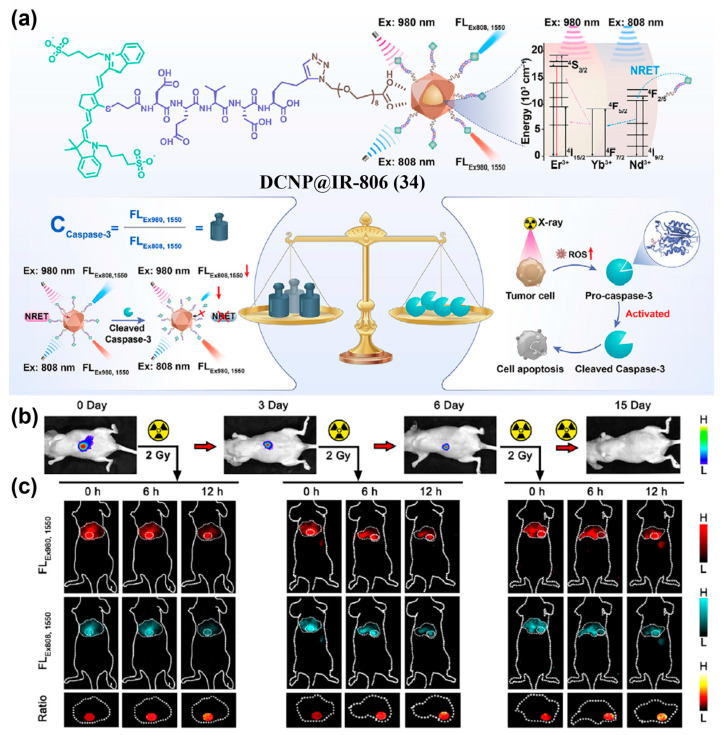
(**a**) Schematic illustration of the ratiometric NIR-II fluorescence nanoplatform DCNP@IR 806 (**34**) for assessing radiotherapy effect based on accurate quantification of activated caspase-3. (**b**) Bioluminescence imaging of OHCC-bearing mouse at different time points after treatment. (**c**) NIR-II fluorescence and ratiometric imaging of OHCC-bearing mice after treatment with **34** and irradiation with X-ray doses. Reproduced from ref. [210] Copyright (2025), with permission from Wiley.

**Figure 42 sensors-25-07080-f042:**
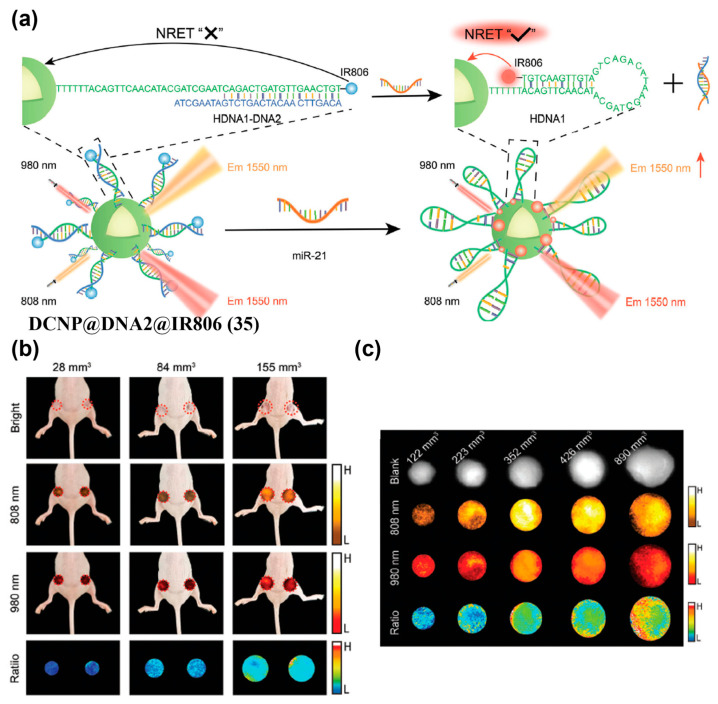
(**a**) Schematic illustration showing the ratiometric NIR-II fluorescence detection of miR-21 using the DCNP@DNA2@IR806 (**35**) nanoprobe based on NRET effect. (**b**) NIR-II fluorescence and ratiometric NIR-II fluorescence images of 4T1 tumor-bearing mice at 18 h after i.v. injection with **35**. (**c**) NIR-II fluorescence and ratiometric NIR-II fluorescence images for the different sized tumors (ex vivo) at 18 h post-injection with **35**. Reproduced from ref. [211] Copyright (2024), with permission from Wiley.

**Table 1 sensors-25-07080-t001:** The photophysical properties of representative NIR-II organic small-molecule fluorophores.

Fluorophore Class	Representative Example	λ_abs_ (nm)	λ_em_ (nm)	Solvent	QY (%)	Ref.
D-A-D fluorophores	CH1055	750	1055	H_2_O	0.03 ^[a]^	[68]
	H2a-4T	738	1024	H_2_O	0.01 ^[a]^	[69]
Cyanine fluorophores	FD1080	1064	1080	H_2_O	0.31 ^[a]^	[70]
	Flav 7	1026	1045	H_2_O	0.53 ^[a]^	[71]
BODIPY fluorophores	NJ1060	910	1060	H_2_O	0.50 ^[a]^	[72]
	WH-4	960	1205	H_2_O	0.05 ^[b]^	[73]
Xanthene fluorophores	CX-3	1089	1140	CHCl_3_	0.09 ^[a]^	[74]
	VIX-4	1014	1210	DCM	0.03 ^[a]^	[75]

^[a]^ The fluorescence QY was calculated relative to IR-26 (0.05% in DCE), ^[b]^ The fluorescent QY was calculated relative to IR-1061 (1.7% in DCM).

## Data Availability

Not applicable.

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
