# Peer review of "Near-Infrared-II Fluorescence Imaging of Tumors with Organic Small-Molecule Fluorophores"

_sensors, 2025, doi:10.3390/s25227080_

Round 1
Reviewer 1 Report
Comments and Suggestions for Authors
This manuscript summarized the latest progress in NIR-II fluorescence probes based on organic small-molecule fluorophores for tumor imaging, focusing on their structural features, design principles of NIR-II fluorescence probes, and applications in tumor imaging. This is a comprehensive and timely review on NIR-II imaging assisted by organic fluorophores, which included “always on” and “activatable” NIR-II probes. The whole review was organized logically and well written. It should be accepted for publication after small changes.
- Some other organic fluorophores such as conjugated polymer nanoparticles, have also shown excellent optical properties and biocompatibility for in vivo imaging. Authors should briefly discuss this in the “Introduction” part.
- For in vivo fluorescence imaging, the signal brightness of the targeted lesion is crucial for imaging SNR. So, authors should discuss some currently used strategies to improve the NIR-II fluorescence brightness of organic small-molecules.
Reviewer 2 Report
Comments and Suggestions for Authors
The manuscript is a high-quality, comprehensive review on NIR-II fluorescence imaging of tumors. The authors demonstrate strong command of the subject, present the material in a logical and accessible manner, and supply informative schematics and examples.
To increase clarity and practical utility, I suggest several modest revisions.
First, when discussing the advantages of NIR-II (reduced scattering and low autofluorescence), include a brief caveat about absorption: in particular, water absorption increases in parts of the NIR-II range, and the most favorable compromise between reduced scattering and minimal absorption. It is also logical to include a graph with the wavelength dependences of the absorption and scattering coefficients of the main tissue chromophores.
Second, consider adding a compact table/sprectra with key spectral parameters of the fluorophores discussed (λabs, λem, εmax, fluorescence quantum yield).
Third, although the article is not claimed as a systematic review, a short statement on the literature selection strategy (databases searched, date range, keywords, inclusion/exclusion criteria) would improve transparency and help readers assess coverage.
On presentation, please add a consolidated List of Abbreviations and use a consistent typographic convention for terms such as in vivo / in vitro (italicized is customary). In Figure 5 correct “Aways-on”; in panel (b)(ii) clarify labels to distinguish “double response” from “comparison with internal reference” and add FL1/FL2 arrows for readability. Also standardize the abbreviation NRET (avoid NRE), correct “inovarian tumors” - “in ovarian tumors”, and fix “fluoresence” - “fluorescence” in the figure 44 caption.
No substantive scientific errors were identified. Following these minor editorial and clarifying revisions, the manuscript is suitable for publication and will be a valuable contribution to the NIR-II biomedical imaging literature. I recommend acceptance pending minor revisions.
Reviewer 3 Report
Comments and Suggestions for Authors
The Review article entitled Near-Infrared-II Fluorescence Imaging of Tumors with Organic Small-Molecule Fluorophores describes recent advances in NIR-II fluorescent probes based on small-molecule organic fluorophores for tumor imaging, highlighting their structural features, design principles, and applications. The authors begin by introducing the chemical structures of fluorophores, followed by a discussion of design strategies, including the distinction between “always-on” and “activatable” fluorescence imaging approaches. Section 4 then presents numerous examples of recent advancements in tumor imaging. The article concludes with a constructive outlook and a clear perspective on future directions. Overall, the review is well executed, with strong attention to detail and accurate referencing.
Minor notes to be addressed:
Line 19: delete “will”
Line 25: change “fluorescence probe” to “fluorescent probe”
Line 64: define SNR
Line 189: change “Fluorescence” to “Fluorescent”
Line 192: change “fluorescence probes” to “fluorescent probes”
Line 197: SNR abbreviation has already been introduced in line 64
Line 201: define “FL” abbreviation in Figure caption
Lines 263-1071 (section 4): Bold every compound number (1-39) throughout the section, as well as in all figure captions (Figures 6-45). This will improve readability and make them more clearly distinguishable from the rest of the text.
Line 545: italicize stereochemical descriptors (S and L)
Line 631: change “quenches” to “quench”
Line 651: RSS abbreviation has already been explained in line 241; change to “RSS (for example, H2S, glutathione (GSH))”
Lines 892-895: Use consistent nomenclature for the Channel 1 and 2 abbreviations with respect to spacing; for example, 'Ch. 1/Ch. 2' versus 'Ch.1/Ch.2'."
Line 995: change “diactetylene-ene” to “diene”
Line 997: change “Hydroxyl” to “hydroxyl”
Reviewer 4 Report
Comments and Suggestions for Authors
The authors summarized the progress in NIR-II organic small-molecule fluorophores for tumor imaging. The work is interesting but modification should be made before publication. Comments:
- The previous review papers about NIR-II fluorescence probes of organic small-molecule fluorophores for tumor imaging should be introduced. The difference of them with this review should be discussed.
- Why Part 3 and Part 4 were not integrated together?
- The advantages and disadvantages of each type of probes mentioned in this review should be discussed and compared.
- The targets of the probes in tumor cells should be provided.
- “Near-infrared-I imaging has achieved meaningful advances in preclinical research and selected clinical applications”. How about near-infrared-II in clinical applications?
Round 2
Reviewer 4 Report
Comments and Suggestions for Authors
Accept in present form